# Indoor Positioning Systems as Critical Infrastructure: An Assessment for Enhanced Location-Based Services

**DOI:** 10.3390/s25164914

**Published:** 2025-08-08

**Authors:** Tesfay Gidey Hailu, Xiansheng Guo, Haonan Si

**Affiliations:** 1Department of Information and Communication Engineering, Addis Ababa Science and Technology University, Addis Ababa 16417, Ethiopia; tesfaygidey21@gmail.com; 2Department of Electronic Engineering, University of Electronic Science and Technology of China, Chengdu 611731, China; sihaonan@std.uestc.edu.cn

**Keywords:** indoor localization, fingerprint-based positioning, critical infrastructure, evaluation metrics, multidimensional metrics framework, dynamic environments

## Abstract

As the demand for context-aware services in smart environments continues to rise, Indoor Positioning Systems (IPSs) have evolved from auxiliary technologies into indispensable components of mission-critical infrastructure. This paper presents a comprehensive, multidimensional evaluation of IPSs through the lens of critical infrastructure, addressing both their technical capabilities and operational limitations across dynamic indoor environments. A structured taxonomy of IPS technologies is developed based on sensing modalities, signal processing techniques, and system architectures. Through an in-depth trade-off analysis, the study highlights the inherent tensions between accuracy, energy efficiency, scalability, and deployment cost—revealing that no single technology meets all performance criteria across application domains. A novel evaluation framework is introduced that integrates traditional performance metrics with emerging requirements such as system resilience, interoperability, and ethical considerations. Empirical results from long-term Wi-Fi fingerprinting experiments demonstrate the impact of temporal signal fluctuations, heterogeneity features, and environmental dynamics on localization accuracy. The proposed adaptive algorithm consistently outperforms baseline models in terms of Mean Absolute Error (MAE) and Root Mean Square Error (RMSE), confirming its robustness under evolving conditions. Furthermore, the paper explores the role of collaborative and infrastructure-free positioning systems as a pathway to achieving scalable and resilient localization in healthcare, logistics, and emergency services. Key challenges including privacy, standardization, and real-world adaptability are identified, and future research directions are proposed to guide the development of context-aware, interoperable, and secure IPS architectures. By reframing IPSs as foundational infrastructure, this work provides a critical roadmap for designing next-generation indoor localization systems that are technically robust, operationally viable, and ethically grounded.

## 1. Introduction

In recent years, Indoor Positioning Systems (IPSs) have become foundational to Location-Based Services (LBSs), significantly enhancing the functionality of smart environments such as hospitals, factories, airports, and commercial buildings [1,2]. IPSs enable real-time tracking, navigation, asset management, and situational awareness in indoor settings where Global Positioning System (GPS) signals are unreliable or unavailable due to structural interference and signal multipath effects [3,4] as visually depicted in Figure 1a,b. For instance, particle-filter-based systems integrated with floor plans can achieve sub-meter accuracy in GPS-denied environments [5], while visible light communication (VLC) systems exploit existing LED infrastructure to attain centimeter-level precision in healthcare settings [6].

Traditionally viewed as auxiliary or convenience-enhancing technologies, IPSs are now increasingly recognized as essential components in environments that demand high precision, reliability, and responsiveness [7]. This shift is particularly evident in sectors such as healthcare, where IPS support the real-time tracking of medical assets and personnel to enhance patient safety and operational efficiency [8], and in industrial automation, where accurate localization is vital for optimizing workflows [9]. Reinforcing this shift, the U.S. Department of Homeland Security has classified resilient Positioning, Navigation, and Timing (PNT) systems—including IPSs—as critical infrastructure (CI) under Executive Order 13905, highlighting their strategic relevance for emergency services, utilities, and transportation [10]. The growing recognition of IPSs as CI is driven by their capacity to reduce the risks associated with operational disruptions that could result in substantial human, economic, or logistical losses. In logistics, for example, IPS-enabled tracking systems have reduced inventory misplacement by 21%, directly contributing to supply chain resilience [11]. In construction and industrial domains, IPSs enhance worker safety and support autonomous operations in hazardous environments, where failure could result in severe consequences [12]. Technological advances such as hybrid Visible Light Positioning (VLP) and Inertial Navigation Systems (INS) now enable centimeter-level accuracy in dynamic and obstructed environments, reinforcing the role of IPSs in robotics and industrial automation [13]. These developments are reflected in global market trends, with the IPS sector projected to grow at a compound annual growth rate (CAGR) of 42.8% by 2030, driven by demand from safety-critical industries [14].

Nevertheless, implementing IPSs in CI applications involves complex trade-offs. For example, Ultra-Wideband (UWB) systems offer high accuracy but entail greater deployment costs and power consumption than Bluetooth-based alternatives [15]. Vision-based systems may deliver fine-grained positioning but pose challenges related to computational load and privacy. As a result, selecting an appropriate IPS solution requires a multi-criteria trade-off analysis that considers factors such as accuracy, latency, energy efficiency, scalability, interoperability, and cost in relation to the specific requirements of CI environments. Despite a substantial body of literature on indoor localization, relatively few studies address IPSs from a critical infrastructure perspective. Most reviews focus on technical classifications or algorithmic developments without fully considering broader concerns such as system resilience, security, interoperability, and regulatory compliance [16,17]. Bridging this gap is essential, particularly as IPSs increasingly underpin operations that require location data to be both accurate and trustworthy under constrained or dynamic conditions. This paper aims to address this research gap by systematically evaluating the current state of IPS technologies with an emphasis on their potential role as critical infrastructure. It introduces a multi-criteria framework for analyzing the trade-offs among different IPS solutions, examining their technical attributes and operational implications. In addition, it identifies unresolved technical challenges, regulatory barriers, and limitations in existing research that hinder the broader adoption of IPSs in mission-critical applications. The analysis draws on a comprehensive review of recent academic and industrial literature spanning diverse use cases, technologies, and performance benchmarks. The main objectives of this paper are fourfold: (1) to conceptualize IPSs as a category of critical infrastructure by articulating their roles, dependencies, and significance in safety-critical domains; (2) to present a structured taxonomy of IPS technologies based on sensing modalities, system architectures, and application contexts; (3) to develop and apply a formal trade-off analysis framework to assess tensions among key system attributes; and (4) to synthesize open research challenges and propose directions for developing scalable, robust, and standards-compliant IPSs. The key contributions of this paper are as follows:-It reframes IPSs from auxiliary technologies to core infrastructure components.-It provides a comprehensive, up-to-date review of IPSs from a CI perspective.-It introduces a structured framework for evaluating trade-offs among IPS technologies.-It identifies critical research gaps, including interoperability, data privacy, and performance under adverse conditions.-It proposes a research agenda to guide the development of resilient and context-aware IPS solutions.

The remainder of this paper is organized as follows: Section 2 presents a comprehensive overview of the background and foundational concepts of IPSs, including taxonomy, enabling technologies, and associated challenges. Section 3 introduces the methodology and experimental framework used to evaluate the performance of IPS techniques under varying indoor conditions. Section 4 outlines the key evaluation criteria for IPSs and provides a comparative trade-off analysis, including a focused discussion on the limitations of GPS in indoor settings. Section 5 offers a critical discussion and synthesis of findings, while Section 6 concludes the paper and outlines future research directions.

## 2. Overview of IPSs

IPSs are advanced technologies that facilitate the tracking and locating of individuals and objects within indoor environments. The adoption of IPSs has surged in recent years due to their versatility across various applications, including asset tracking, navigation, safety and security, and retail analytics [18,19]. The literature categorizes IPSs using several approaches:

System Topology: This classification reflects the hardware and software architecture of the IPSs. The primary system topologies include network-based, terminal-based, and terminal-assisted configurations [20,21,22,23].

Positioning Technology: IPSs can be distinguished based on the technologies employed for location determination. Common techniques include Wi-Fi, Bluetooth, RFID, UWB, geomagnetic sensing, and inertial navigation [18,19,24,25].

Signal Measurement Methods: Various methodologies are utilized to measure signals from transmitting devices in indoor settings. Common approaches include time-based measurements (Time of Arrival, TDOA), angle-based measurements (Angle of Arrival), and Received Signal Strength Indicator (RSSI) techniques [18,19].

Indoor Navigation Systems: Some classifications focus on how IPS technologies integrate into broader indoor navigation systems, enhancing user experience and operational efficiency [26,27,28].

IPSs are further categorized into infrastructure-based and infrastructure-free systems. Infrastructure-based IPSs depend on fixed beacons or sensors, while infrastructure-free systems utilize sensors integrated into mobile devices. Infrastructure-based systems can employ Wi-Fi, BLE, or RFID technologies, whereas infrastructure-free systems may utilize Inertial Measurement Units (IMUs) or magnetic field-based positioning [18]. In terms of coverage, location systems can be divided into three categories: (i) indoor location systems, (ii) wide-area location systems (based on cellular networks), and (iii) global location systems. Additionally, indoor positioning technologies are classified into two groups based on signal type: (i) radio-based signals (such as Wi-Fi, RFID, Bluetooth, Zigbee, and UWB) and (ii) non-radio-based signals (including ultrasound, infrared, and geomagnetic fields).

IPSs employ a variety of algorithmic strategies and architectural models to estimate the location of devices within indoor environments. Broadly, three primary algorithmic categories are widely recognized in the literature: (i) triangulation, (ii) proximity-based estimation, and (iii) scene analysis. Each category leverages distinct signal characteristics and modeling techniques to infer position. The triangulation method utilizes geometric principles to calculate the target’s position relative to multiple known reference points. It comprises two core techniques: *lateration*, which estimates distances based on signal propagation metrics (e.g., TOA, RSSI), and *angulation*, which relies on directional information (e.g., AOA) to calculate angular displacement. While triangulation can achieve reasonable accuracy under line-of-sight (LOS) conditions, it suffers from performance degradation in multipath-prone or obstructed environments.

In contrast, proximity-based estimation determines the target’s location by associating it with the nearest known transmitter—such as a Bluetooth beacon, RFID tag, or Wi-Fi access point—based on the strongest received signal. This approach is computationally efficient and commonly used in commercial deployments. However, it typically offers only coarse-grained accuracy, especially in environments where signal strength fluctuates due to interference or spatial ambiguity [1,3,21,29,30]. The third category, scene analysis, includes fingerprinting-based localization techniques, which have shown promise in complex indoor spaces. Scene analysis operates in two phases: a training (offline) phase, in which signal measurements—such as RSSI values—are collected at predefined reference points and stored in a structured database; and a testing (online) phase, in which real-time measurements are compared against the database using algorithms such as k-nearest neighbors (k-NN), support vector machines (SVM), or neural networks to estimate position. While scene analysis can achieve high accuracy in non-line-of-sight (NLOS) environments, it requires extensive data collection and periodic recalibration.

Figure 2 illustrates the core dimensions and taxonomic structure used in the design, classification, and evaluation of IPS technologies. This hierarchical framework encompasses three major classification criteria: system topology, positioning technologies, and signal measurement methods. The system topology dimension distinguishes between infrastructure-based systems, which rely on fixed beacons or anchors (e.g., Wi-Fi routers, BLE beacons), and infrastructure-free systems, which typically use inertial sensors or device-to-device collaboration without pre-installed hardware. The positioning technologies dimension identifies common signal transmission platforms, including Wi-Fi, BLE, UWB, RFID, magnetic field sensing, acoustic signals, and VLP. The signal measurement methods dimension categorizes the types of measurements used to extract spatial information, such as RSSI, CSI, TOA, AOA, TDOA, and Doppler shift.

This taxonomic structure serves as a design-oriented lens through which IPS architectures can be systematically analyzed and compared based on their hardware dependencies, signal processing strategies, and environmental adaptability. The scene analysis approach is further detailed through Figure 3, which illustrates a typical Wi-Fi fingerprinting process. During the training phase, RSSI measurements are collected from multiple access points (AP1, AP2, …, APn) at known reference locations (x1, y1), (x2, y2), …, and stored in a fingerprint database. In the online phase, real-time RSSI values are acquired and matched against the stored fingerprints using a suitable matching algorithm to predict the user’s location. This method leverages the spatial uniqueness of signal patterns in indoor environments to improve localization accuracy. The integration of algorithmic approaches into the proposed taxonomic framework enables a structured evaluation and design of IPS technologies based on three core objectives: ubiquity, which emphasizes compatibility with existing infrastructure and wide coverage; seamlessness, which promotes interoperability and adaptive functioning across heterogeneous environments; and intelligence, which supports the use of data-driven and learning-based models to enhance positioning performance under uncertainty. The classification framework outlined in Figure 2 thus provides a comprehensive basis for evaluating IPSs in terms of their scalability, precision, and context-awareness in dynamic indoor environments.

### 2.1. Applications of IPSs

With the advance of location-based services, location applications have attracted a lot of attention. Despite the GPS is considered as a standard promising method for outdoor positioning its applications have been restricted in indoor localization because of the unattainability of line-of-sight path (LOS). Many location systems are an important issue in LBSs, which are used to track or navigate the user’s location. There are triple classifications of location systems, in general, known as (i) indoor location systems (ii) wide area location systems (based cellular networks) and (iii) global location systems. Moreover, depending on the use of signals the indoor positioning technologies are classified into two basic groups: (i) radio-based signals and (ii) non-radio-based signals. For example, there are several indoor location technologies based on radio signals, which include: Wi-Fi, RFID, Bluetooth, Zigbee, and UWB, while ultrasound, infrared, and geomagnetic fields are some examples of location technologies based on non-radio signals.

(a)Public building

IPSs have emerged as critical enablers of navigation, security, and operational efficiency in public buildings, addressing challenges inherent to GPS-denied environments [31]. Early infrastructure-light approaches leveraged Wi-Fi access points (APs) [32] and model-based techniques [33,34] to minimize deployment costs while achieving scalability. These foundational efforts laid the groundwork for hybrid systems, such as the Wi-Fi-Vision (Wi-Vi) fusion by Huang et al. [35], which integrates Wi-Fi signals with visual EXIT sign landmarks to achieve sub-meter localization accuracy (<0.5 m) and 95–98% site recognition rates in large-scale public spaces. Complementing this, Schäfer et al. [36] demonstrated the viability of CAD-driven semantic mapping for real-time particle-filter localization, achieving 95% structural detection accuracy with sub-150 ms processing speeds, thereby eliminating manual map creation.

For emergency scenarios, advancements include Bluetooth-based evacuation systems [37] employing Kriging interpolation (2.5 m precision) and vibration-sensing techniques [38] that achieve sub-meter accuracy through footstep-induced floor vibrations, ensuring reliable indoor-outdoor navigation and evacuation. Meanwhile, Building Information Modeling (BIM)-driven frameworks [39] support complex indoor navigation and localization through semantic mapping and pre-rendered 3D visual interfaces, enhancing user guidance in non-convex spatial layouts. Recent innovations emphasize multi-sensor integration and context-aware adaptability. For instance, Czogalla and Naumann [40] combined BLE beacons, pedestrian dead reckoning (PDR), and topological floor models to enable real-time navigation across multi-level facilities. Alarifi et al. [41] highlighted Ultra-Wideband (UWB) as a resilient solution for multipath-prone environments, while Poston [38] showcased device-free localization via seismic signal processing. These technologies support diverse applications, from personalized services for visually impaired users [42,43] to real-time foot traffic analytics [44] for optimizing operations in high-density venues such as airports and hospitals. These advancements therefore underscore the transformative potential of IPSs in public infrastructure. By harmonizing infrastructure-light designs [42,44], multi-sensor fusion [35,36] and semantic context-awareness [39], modern IPSs achieve robust accuracy, scalability, and real-time performance. Their applications span improved wayfinding [45,46,47] enhanced security [48], emergency response [49], and operational management [45], positioning IPSs as indispensable tools for smart, user-centric public environments.

(b)Indoor navigation and Tracking objects

IPSs have emerged as critical solutions for navigation and object tracking in environments where GPS signals are unreliable or unavailable [50]. By leveraging technologies such as Wi-Fi, BLE, UWB, and sensor fusion, IPSs enable real-time localization with sub-meter accuracy, addressing challenges posed by complex indoor layouts and dynamic obstacles [51,52]. Recent advancements in machine learning and hybrid algorithms have further enhanced robustness against signal interference, enabling adaptive tracking of both static and mobile objects [53,54]. For instance, fingerprinting-based methods using Channel State Information (CSI) and inertial measurement units (IMUs) have demonstrated improved precision in crowded spaces like hospitals and airports [55]. Moreover, IPSs facilitate applications ranging from asset management in warehouses to emergency evacuation guidance, with emerging frameworks integrating 6G and IoT for ultra-low latency communication [56,57]. Despite progress, challenges such as scalability, energy efficiency, and privacy preservation remain focal points for future research [51,58]. In addition, IPSSs have a wide range of applications in indoor navigation and tracking objects [59]. The use of IPSSs has been studied for assisting visually impaired people and improving accessibility in various industries [60,61]. Esri’s ArcGIS IPSs and Mapsted’s hardware-free indoor location tracking technology are examples of IPSs that have been developed and applied in various settings [62,63]. The research highlights the need for infrastructure-free indoor navigation systems and collaborative indoor positioning systems to improve indoor location accuracy and accessibility [64]. The applications of these IPSSs include commercial, military, retail, inventory tracking, and more [65].

IPSs leveraging Wi-Fi, BLE, and UWB technologies have emerged as robust solutions for real-time localization and wayfinding in complex indoor environments. UWB systems, with their centimeter-level accuracy and resilience to multipath interference, are increasingly adopted for industrial automation and augmented reality applications, achieving precision below 10 cm through advanced time-of-flight measurements [66]. Meanwhile, BLE 5.1+ systems utilize AoA and mesh networking capabilities to deliver scalable, cost-effective navigation in large public spaces such as airports and shopping malls [67]. Wi-Fi 6E further enhances positioning performance by exploiting the 6 GHz band’s wider channels and reduced congestion, enabling high-density tracking in crowded venues like stadiums with sub-meter accuracy [68]. Recent advancements integrate machine learning (ML) to optimize fingerprinting databases and mitigate environmental dynamics, such as moving obstacles and signal fluctuations, improving robustness in hybrid systems that fuse Wi-Fi/BLE with inertial sensors [69]. For instance, hospitals deploy UWB-enabled asset tracking to monitor medical equipment, while retail environments use BLE beacons for personalized customer navigation [66,67]. These technologies not only address scalability and infrastructure costs but also align with emerging standards like IEEE 802.15.4z for UWB, ensuring interoperability and future-proofing deployments [70]. As IPSs evolve, the synergy of AI-driven analytics and multi-technology fusion promises to redefine indoor navigation, offering seamless user experiences across healthcare, logistics, and smart cities. In conclusion, no single IPS technology universally addresses all indoor navigation and tracking needs. UWB and IR systems suit high-precision industrial applications, while WLAN and RFID offer cost-effective solutions for large-scale environments. Future advancements in hybrid systems and AI-driven algorithms will enhance robustness and scalability, enabling broader adoption across healthcare, logistics, and smart cities.

(c)Healthcare

IPSs have emerged as pivotal tools for addressing critical challenges in healthcare, including workflow inefficiencies, medical errors, and operational costs. Broad surveys by Nia et al. [71] and Jeny et al. [72] contextualize healthcare applications within the broader evolution of IPS technologies. Nia et al. [71] emphasized energy-efficient wireless personal area networks, a critical priority for wearable healthcare devices requiring prolonged operation. Complementing this, Jeny et al. [72] highlighted advancements in artificial intelligence (AI) and adaptive signal processing, demonstrating their potential to enhance localization accuracy in dynamic hospital environments through real-time noise filtering and multipath mitigation. These foundational studies frame the dual challenges of energy efficiency and precision inherent to healthcare IPS design. Building on this, Bazo et al. [73] conducted a systematic survey of Real-Time Location Systems (RTLS) in healthcare, identifying barriers such as sensor limitations, infrastructure costs, and human resistance to adoption. Their taxonomy categorizes hardware, application, and maintenance requirements for successful deployments, advocating for hybrid solutions that balance scalability, accuracy, and power consumption [73].

Recent technical innovations have sought to address these challenges. Wyffels J et al. [74] prioritized scalability by proposing a distributed, RSS-based IPSs leveraging existing nurse call infrastructure. Their system achieved room-level accuracy at minimal cost but lacked the precision required for critical applications like surgical navigation [74]. To overcome resolution limitations, Nguyen QH et al. [75] developed a BLE iBeacon system with an AI-enhanced Least Square Estimation (LSE) method, achieving 12 cm accuracy in controlled settings. However, dense beacon deployment and battery dependence hindered scalability [75], while Neburka et al. [76] empirically validated BLE’s environmental sensitivity, revealing RSSI fluctuations of 6 dBm in real-world corridors due to multipath interference and WLAN coexistence [76]. Energy efficiency was further prioritized by Kanan and Elhassan [77], who introduced a hybrid IPSS combining batteryless EnOcean patient calls with Wi-Fi ToA and AoA. Their IoT-enabled system eliminated battery maintenance but achieved only 5 m accuracy, underscoring trade-offs between sustainability and precision [77]. Contrasting infrastructure-dependent approaches, Vidal et al. [78] demonstrated a smartphone-based Pedestrian Dead Reckoning (PDR) system using accelerometers and gyroscopes. With 89.93% step detection accuracy, their infrastructure-free solution highlighted the potential of wearable sensors but faced cumulative errors (4.5% over 150 steps) and calibration dependencies [78].

The convergence of IPSs with IoT and ambient intelligence has expanded their healthcare applicability. Sousa et al. [79] validated smartphone-based activity recognition for applications like fall detection and elderly care, achieving robust classification of daily activities through inertial sensors. However, their reliance on device-specific calibration limited generalizability across heterogeneous hardware [79]. Scrivano A et al. [80] systematized healthcare IoT integration, classifying IPS applications into domains such as infection control, telehealth, and indoor navigation. Their review advocated for context-aware systems but identified unresolved challenges in latency, energy trade-offs, and interoperability across proprietary platforms [80]. Finally, Acampora G et al. [81] proposed a framework integrating environmental sensing with location data, offering a blueprint for ambient intelligence in healthcare. For instance, correlating spatial data with air-quality sensors could enable adaptive interventions, such as adjusting ventilation in high-risk infection zones or triggering alerts for patients entering restricted areas [81].

Together, these studies affirm the transformative potential of IPSs in healthcare while exposing critical gaps: (1) Precision-Cost Trade-offs: High accuracy (e.g., AI-enhanced systems [72,75]) demands costly infrastructure or computational resources, limiting accessibility for resource-constrained settings; (2) Energy Efficiency vs. Performance: Battery-free designs [77] and sensor-only tracking [78] sacrifice precision or scalability; (3) Interoperability: Fragmented standards hinder integration with legacy healthcare systems [73,80]; and (4) Ethical Concerns: Continuous tracking raises patient privacy issues, particularly in mental health and elderly care [80]. The convergence of AI-driven localization, wearable sensors, and IoT-enabled ambient intelligence presents a promising yet unrealized vision for holistic healthcare IPS solutions. Future work must prioritize hybrid architectures that harmonize infrastructure efficiency, real-time adaptability, and ethical safeguards to bridge these gaps.

(d)Manufacturing industries

Recent advancements in IPSSs have significantly enhanced operational efficiency across various industries, particularly in logistics and manufacturing. The review by Zhu et al. emphasizes the increasing relevance of IPSs in logistics, addressing the need for safety enhancement and effective material flow control in diverse environments such as industrial facilities and hospitals. The study systematically evaluates various IPS technologies, with a focus on radio-based solutions like UWB, RFID, and BLE. The authors conclude that hybrid approaches integrating multiple technologies are essential for achieving high accuracy and reliability, especially in complex operational contexts [82]. In exploring inventory management challenges, Octaviani et al. investigate the application of BLE beacons in warehouses. Their findings reveal that BLE technology facilitates effective zone mapping and indoor navigation, significantly improving employee productivity. The study underscores the transformative potential of BLE beacons in enhancing inventory placement strategies by providing real-time notifications and interactive mapping capabilities [83]. A comprehensive survey by Rahman et al. highlights the necessity of IPSs in IoT-based applications, particularly in scenarios where GPS coverage is inadequate. The authors provide an overview of the latest positioning methods and technologies, discussing the challenges of integrating IPSs within IoT ecosystems. Their conclusions advocate for a hybrid system that combines various communication technologies to improve performance and accuracy, addressing the growing demand for context-aware services in smart applications [84].

Subakti et al. propose a markerless AR system that utilizes deep learning to enhance visualization and interaction with machines in smart factories. Their prototype demonstrates high detection accuracy and interactive features, suggesting that integrating this AR system with indoor localization technologies will further enhance its functionality and user experience [85]. Zhu et al. also present a particle filter-based localization method using leaky coaxial cables (LCX) to overcome the limitations of conventional antennas in industrial settings. By employing time difference of arrival (TDOA) measurements, they achieve superior localization accuracy, making LCX a promising solution for advanced manufacturing applications [86]. Future directions on enhanced positioning services by Kuhlmann et al. emphasize the importance of integrating various data sources to enhance indoor positioning accuracy in smart factories. Their proposed framework aims to provide real-time localization services while addressing integration challenges, illustrating its application through a use case involving automated guided vehicles (AGVs) [87]. The innovative study by Maly et al. introduces a laser scanning system for indoor localization, capable of tracking multiple mobile tags with high precision. The authors demonstrate that their system achieves sub-millimeter resolution and emphasize the potential for further enhancements, including 3D localization capabilities, to improve its applicability in dynamic environments [88].

In a real-time application, the research by Koller et al. presents a multi-channel UWB positioning system designed for industrial scenarios. Their findings indicate that using multiple transceivers significantly improves localization accuracy, making the system suitable for critical applications in manufacturing and logistics [89]. The application of IPSs in optical lens manufacturing is explored by Rahman et al., who utilize BLE sensors for signal strength fingerprinting. Their results show that the designed system can achieve reliable positioning, although challenges remain in complex environments. Future work will focus on optimizing configurations to minimize signal interference and improve accuracy [90].

In the context of indoor exhibitions, Wang et al. investigate the use of BLE beacons to enhance visitor navigation and engagement. Their study highlights the potential for further applications of indoor positioning technologies, emphasizing the need for improved accuracy and integration with other systems to enrich user experiences [91]. Lastly, the study by Chatterjee et al. proposes a UWB-based asset tracking architecture for Industry 4.0, addressing the need for effective asset management in industrial environments. The authors integrate triangulation methods to enhance tracking precision and propose future work that incorporates machine learning techniques for error mitigation [92]. Together, these studies underscore the critical role of innovative indoor positioning technologies in advancing operational efficiency across various sectors. The integration of IPSs with AR, IoT, and advanced tracking methods illustrates a comprehensive approach to addressing the challenges of indoor localization, paving the way for enhanced automation and productivity in smart environments.

(e)Public Security

The evolution of IPSs have been driven by diverse technological and methodological advancements, each addressing critical challenges such as accuracy, privacy, scalability, and environmental adaptability. Early foundational work by [93] systematically reviewed IPS technologies (e.g., Wi-Fi, Bluetooth, RFID), evaluating their cost, accuracy, and security risks. This study highlighted the absence of standardized frameworks and the urgent need for privacy safeguards, given the sensitivity of location-based data. Building on this, Ref. [94] proposed a scene classification approach to optimize IPS deployment across diverse architectural environments (e.g., hospitals), tailoring algorithms to specific indoor scenarios. By categorizing environments into three classes and refining wireless positioning methods, Ref. [94] demonstrated improved adaptability, emphasizing the limitations of single-technology solutions and advocating for hybrid systems. To address data collection inefficiencies in visual-based IPSs, Ref. [95] introduced a video stream-based method for rapid map-building, replacing labor-intensive image databases. While achieving 70% localization accuracy within 2 m, the approach prioritized scalability over precision, suggesting future integration with laser radar for error reduction. Similarly, Ref. [96] advanced visual localization by combining Speed-Up Robust Features (SURF) with homography matrices, achieving sub-meter accuracy and directional angle estimation (within 8° error). This method reduced storage overhead and latency, making it suitable for real-time navigation applications.

In vehicular contexts, Refs. [97,98] explored fingerprint database filtering and dead reckoning for secure traffic flow management. Ref. [97] utilized RSSI-based Wi-Fi fingerprinting to enhance data transmission accuracy in intelligent transportation systems, while [98] integrated dead reckoning with Wi-Fi and computer vision (CV) to secure vehicle interactions in GPS-denied environments like underground garages. Both studies underscored the importance of fingerprint-matching algorithms (e.g., nearest neighbor, probabilistic) in improving traffic efficiency and safety. For rescue operations, Ref. [99] tackled altitude errors in firefighter navigation systems caused by extreme temperatures. By synthesizing IMU-derived heading and pitch data, their elastic correction method dynamically adjusted altitude measurements, reducing rescue time and enhancing robustness in harsh environments. Privacy and security emerged as recurring themes. Ref. [100] proposed BILPAS, a blockchain-based framework for secure person-finding services, using a three-way Diffie-Hellman key agreement to establish encrypted communication tunnels between users. This approach minimized interaction overhead while ensuring compliance with privacy regulations. Complementing this, Ref. [101] introduced a lightweight group anonymity scheme for Wi-Fi fingerprinting, employing Paillier encryption and false fingerprints to protect multi-user locations. By unifying group queries, the method reduced computational costs while maintaining sub-meter accuracy, balancing privacy and efficiency.

Deep learning advancements further transformed IPSSs. Ref. [102] developed RRIFLoc, a deep residual network that fused RSSI, signal strength difference (SSD), and kurtosis into radio robust image fingerprints (RRIF), reducing localization errors by 56.87% compared to state-of-the-art methods. Meanwhile, Ref. [103] proposed SPORT, a signal propagation-based outlier reduction technique that corrected RSS inconsistencies in crowdsourced datasets, smoothing radio maps and improving accuracy in both offline and online phases. Finally, application-centric innovations like [104]’s AR GuideX system combined augmented reality overlays with Bluetooth beacons and Wi-Fi to deliver immersive navigation.

(f)Monitoring People and Activities

Recent advancements in indoor positioning and activity monitoring systems have significantly enhanced healthcare, elderly care, and emergency response applications. Vision-based approaches, such as the low-cost Kinect depth camera system by [105], enable non-invasive multi-user tracking in smart homes through optimized calibration and data fusion, with future aims to integrate identity recognition for personalized activity monitoring. Complementing this, Ref. [106] leverages edge computing and privacy-preserving camera networks to achieve 1.41 m localization accuracy and 88.6% multi-object tracking performance in large indoor spaces (1700 m^2^), emphasizing scalability for crowded environments. Device-free localization (DFL) methods, like the RF signal absorption/reflection technique by [107], eliminate wearable dependencies, achieving 1.5 m resolution for elderly care, while [108] employs BLE beacons to track multiple individuals in day care centers via RSSI analysis, reducing caregiver burden. For assistive navigation, Ref. [109] introduces a Li-Fi-based system using LED light transmission and ultrasonic sensors to guide visually impaired users, overcoming Wi-Fi/RFID limitations. Wearable solutions also show promise: Ref. [110] combines smartwatch RSSI fingerprinting with SLAM calibration for real-time tracking, while [111] integrates IR beacons and LoRaWAN for room-level localization and step detection in mild cognitive impairment (MCI) patients, achieving > 90% accuracy. Child safety is addressed by [112] through an IoT bracelet with indoor localization and health monitoring, whereas [113]’s rescue wearable (JARVIS) employs IMU sensors and PDR algorithms for 50 cm-accurate emergency responder tracking. Novel sensor innovations, such as [114]’s air pressure-based step detection (7.5% error), reduce reliance on accelerometers. Collaborative activity analysis is enabled by [115]’s synchronized cardiac-GPS data fusion, revealing behavioral patterns in social tasks. Context-aware emergency detection by [116] integrates sound, activity, and location via deep neural networks on mobile devices, achieving high recognition accuracy. In nursing homes, Ref. [117]’s ACTIVA system uses fog-cloud architecture and fuzzy logic for real-time anomaly alerts, validated through stakeholder feedback. Finally, Ref. [118] proposes a holistic IoT-based AAL system for elderly care, correlating environmental and biometric data to trigger alerts. Generally, these studies highlight the diversification of indoor positioning technologies—spanning vision, RF, Li-Fi, wearables, and edge computing—to address critical needs in healthcare, safety, and autonomy, with ongoing efforts to enhance accuracy, privacy, and real-world applicability [105,106,107,108,109,110,111,112,113,114,115,116,117,118].

(g)Service Industries/Sectors

Recent advancements in indoor localization have explored diverse methodologies to address challenges such as accuracy, scalability, cost, and adaptability to complex environments. VLC emerged as a promising low-cost solution in [119], which proposed using smartphone microphone jacks as optical receivers to decode location IDs from LED light states. This approach eliminated the need for specialized hardware, leveraging ubiquitous smartphones for indoor navigation. The system demonstrated reliability in controlled environments, though its dependency on direct line-of-sight (LOS) limited applicability in obstructed spaces. Building on broader infrastructure needs, [84,120] surveyed IPSs for IoT, emphasizing hybrid frameworks that integrate Bluetooth 5.1, UWB, and VLC. It underscored the role of AI and ML in overcoming scalability and standardization challenges, advocating for unified platforms to support context-aware services.

Reference [121] introduced a real-time Wi-Fi localization system using Deep Neural Networks (DNN) trained on RSSI fingerprints in 3GPP-standardized offices. While achieving 97.5% classification accuracy, the trade-off between processing speed (faster than traditional methods) and positional accuracy highlighted opportunities for model optimization. In addition, Ref. [122] proposed a novel approach to localization, conceptualizing it as a network service (LaaNS) by integrating location information within DNS records. This DNS-based framework enabled global interoperability across heterogeneous systems, with prototype tests confirming low latency and scalability. Sensor fusion took center stage in [123], which integrated Wi-Fi and geomagnetic fingerprints using ANN and KNN algorithms. The hybrid WMLoc system achieved < 2.6 m accuracy in complex buildings, demonstrating robustness against signal fluctuations. Similarly, Ref. [124] tackled non-line-of-sight (NLOS) challenges by transforming CSI into 2D images for DCNN analysis, achieving sub-meter accuracy in simulations. These works underscored the potential of AI-driven fingerprinting in dynamic environments.

For large-area applications, Ref. [125] proposed Fi-Vi, a hierarchical system combining sparse Wi-Fi fingerprinting (FBL) and visual positioning (VBL). By narrowing search areas via coarse wireless regions, computational load reduced by 84%, enabling efficient navigation in expansive spaces like airports. Study in [126] further enhanced autonomy through activity sequence-based localization, using smartphone sensors and Hidden Markov Models (HMM) to map pedestrian movements (e.g., elevator use, turns) to indoor road networks. This achieved 1.3 m accuracy without prior knowledge of starting points, ideal for structured environments. Industrial applications were addressed in [127], which validated UWB’s robustness in metal-rich galvanic facilities. Despite multipath interference, sub-meter accuracy proved its viability for process monitoring in harsh settings [128,129] advanced AoA-based systems: RcLoc optimized antenna orientations on COTS Wi-Fi, achieving 0.4 m median error, while OpArray refined phase calibration and array rotations to reach 0.5 m accuracy. Both highlighted geometric optimization as a critical, yet underexplored, factor in precision.

Finally, research in [130] achieved a breakthrough with Guoguo, an acoustic-based system offering centimeter-level accuracy via inaudible signals. By synchronizing smartphone microphones with anchor networks, it attained 6–15 cm precision, enabling applications like assistive navigation for the visually impaired. These studies collectively illustrate a shift toward hybrid systems, AI/ML integration, and infrastructure-light solutions. While VLC and acoustic methods prioritize cost and precision, Wi-Fi/UWB hybrids balance accuracy with scalability. Future work must address real-world deployment challenges, including dynamic environmental adaptation, energy efficiency, and standardization. The convergence of AI-driven analytics, multi-sensor fusion, and IoT frameworks will likely dominate next-generation IPSSs, bridging the gap between theoretical innovation and practical implementation.

### 2.2. Related Works

IPSs have evolved into critical enablers of LBSs, particularly in GPS-denied environments. Their widespread adoption across domains such as healthcare, logistics, public safety, and industrial automation highlights the growing need for systems that are not only accurate but also **ubiquitous**, **seamless**, and **intelligent**. This section reviews major technological trends in IPSs research through the lens of these three core design objectives, which form the foundation for future-ready and infrastructure-grade indoor localization systems (see Figure 4).

#### 2.2.1. Ubiquity

Ubiquity (Scalable and Infrastructure-Light Deployments) in IPSs refers to the system’s ability to operate reliably across diverse environments with minimal installation overhead by leveraging existing infrastructure. Research aligned with this objective [32,33,34,35,36,37,38,51,58,70,75,76,77,83,86,119] seeks cost-effective, easily deployable solutions that support widespread adoption. Early implementations in public buildings utilized existing Wi-Fi access points and model-based localization techniques to reduce installation costs and facilitate scalability [32,33,34]. These evolved into hybrid models that combine Wi-Fi with vision-based elements like EXIT sign recognition [35], achieving sub-meter accuracy and high site recognition rates. CAD-driven semantic mapping further supported real-time localization without manual calibration [36]. In public safety contexts, Bluetooth-based evacuation frameworks have leveraged Kriging interpolation and vibration sensing for accurate indoor-outdoor transitions [37,38]. Batteryless IPSs offer long-term sustainability but often at the expense of accuracy [77]. VLC systems using LED signals and smartphone microphones provide low-cost infrastructure-light alternatives, though with line-of-sight limitations [119]. BLE beacons have facilitated efficient zone mapping and warehouse management in logistics [83], while Wi-Fi-based IPSs continue to offer scalability and cost-effectiveness, albeit with increased susceptibility to interference [51,58]. UWB systems deliver high accuracy but require costly infrastructure [70].

#### 2.2.2. Seamlessness

Seamlessness (Interoperability and Environmental Continuity) emphasizes uninterrupted IPS functionality across heterogeneous environments, platforms, and contexts. It involves ensuring consistent localization despite environmental variability, technological fragmentation, and user movement across spatial boundaries. To enhance seamless localization, multi-modal sensor fusion has been extensively explored. Integration of BLE, Pedestrian Dead Reckoning (PDR), and Building Information Modeling (BIM) has enabled effective cross-floor navigation in complex facilities [39,40]. BLE systems employing angle-of-arrival (AoA) and mesh networking have extended spatial coverage in large venues [67]. Healthcare systems require seamless interoperability with existing hospital infrastructures, prompting concerns about data standardization, privacy, and latency [73,80]. Blockchain-based IPS frameworks enable secure and anonymous data exchange, supporting integration in security-critical environments [100,101]. In emergency settings, video stream-based mapping and homography estimation techniques allow robust, real-time localization under dynamic conditions [95,96]. DNS-integrated IPSs and pedestrian tracking systems based on global standards ensure interoperability and adaptability across facilities [122,126]. Fingerprint filtering algorithms have improved vehicular localization but face challenges in signal-unstable environments [97,98,99]. These contributions reflect the importance of adaptive and interoperable systems capable of navigating complex and evolving operational conditions.

#### 2.2.3. Intelligence

Intelligence (Data-Driven and Context-Aware Localization) in IPSs refers to adaptive, learning-based systems that leverage data to improve positioning performance amid noise, dynamics, and environmental uncertainties. It reflects the shift toward AI-powered, context-aware localization models. Machine learning and deep learning methods have been extensively applied to RSS fingerprinting, signal regression, and generalization across domains [69,121,123]. Hybrid approaches fusing RSSI, inertial sensors, and vision enable robust context-aware localization [69,124]. AI-enhanced BLE and UWB systems demonstrate improved performance in uncertain settings [66,72,75]. In healthcare, intelligent IPSs facilitate activity monitoring, cognitive assistance, and fall detection. Vision-based and wearable sensor systems, supported by edge computing, offer non-invasive, multi-object tracking [105,106,113]. Despite benefits, calibration drift and hardware heterogeneity challenge consistent performance [79,110,111,114]. Reinforcement learning, semantic localization, and adaptive filtering enhance system robustness and decision-making under complex conditions [97,118]. In industrial settings, deep learning and Augmented reality (AR) interfaces support real-time interaction with machines and environments. Applications include automated guided vehicles (AGVs) tracking, collaborative robot guidance, and predictive analytics [85,87,88,89]. Trade-offs involving latency, cost, and system complexity remain central [82,84]. Finally, RSSI-based neural networks and geomagnetic-Wi-Fi fusion have improved performance in dense signal environments, while hybrid systems integrating visual, RF, and sensor data expand the scope of intelligent IPS architectures [121,123,124,125].

**Figure 4 sensors-25-04914-f004:**
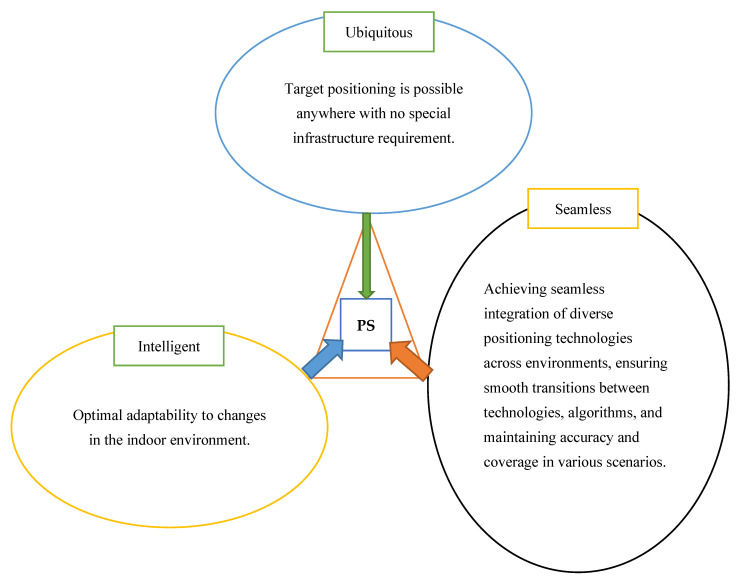
Goal of positioning system. PS: positioning system.

Generally, the state-of-the-art reveals that no single IPS technology or approach universally satisfies the diverse demands of indoor localization across domains. High accuracy often entails high cost and computational complexity, while scalable and low-energy systems may compromise on precision or responsiveness. Privacy and security trade-offs become increasingly critical in sensitive sectors like healthcare and public safety. Therefore, future directions in IPS research must navigate these trade-offs through hybrid, AI-enhanced architectures that harmonize performance, energy efficiency, cost, and ethical considerations. The trajectory of IPS innovation is steadily advancing toward context-aware, resilient, and user-centric positioning services that redefine indoor spatial intelligence as a pillar of smart infrastructure.

### 2.3. Challenges and Trade-Off of IPSs

IPSs are subject to a set of technical and operational challenges that complicate their deployment, scalability, and long-term reliability, particularly when envisioned as critical infrastructure. One of the foremost challenges is the accuracy-environment trade-off: while technologies such as UWB and BLE offer high accuracy, their performance degrades in multipath-rich, dynamic indoor environments due to signal reflection, occlusion, and NLOS conditions [66,69]. Feature heterogeneity and temporal signal variation further undermine the consistency of signal-based fingerprinting approaches, as calibration data may become outdated or misaligned across different hardware platforms [131,132]. Wi-Fi fingerprinting, while cost-effective and widely used, suffers from instability over time and across space, which significantly affects model generalizability [75].

A key trade-off involves accuracy versus energy efficiency. While AI-enhanced sensor fusion and multi-modal localization can improve precision, they often impose significant computational and energy costs, making them less suitable for battery-constrained applications such as wearables or mobile health monitoring systems [71,77]. The scalability–infrastructure trade-off is another major bottleneck. High-precision systems generally require dense anchor deployments, which can be costly and impractical in large or multi-floor environments. Conversely, infrastructure-light systems are more scalable but tend to deliver reduced accuracy or resilience in challenging contexts [68,84].

Privacy and data security present additional challenges, especially in sensitive domains like healthcare and public surveillance. IPS solutions must balance continuous user tracking with robust privacy-preserving mechanisms to comply with data protection regulations and ethical standards [100,101]. Finally, the lack of standardization and interoperability across IPS technologies hampers integration with existing systems and limits the development of universally compatible solutions. As IPSs move toward infrastructure-grade roles, addressing these trade-offs becomes essential to ensure system adaptability, reliability, and trustworthiness across diverse use cases. This paper critically examines the core challenges and operational threats that constrain the widespread deployment of IPSs, emphasizing the absence of a universally accepted standard—akin to GPS for outdoor positioning—as a fundamental barrier to their broad integration into critical infrastructure applications.

Stand-alone Failure System/technologies

As depicted in Figure 5, a wide range of wireless technologies has been explored for indoor positioning, including FM radio, ultrasound, visible light, magnetic fields, Wi-Fi, RFID, BLE, UWB, and Zigbee. Each of these technologies offers distinct benefits and limitations in terms of accuracy, coverage, cost, and susceptibility to environmental factors. However, many existing systems rely on standalone technologies, making them vulnerable to performance degradation or complete failure in the presence of signal attenuation, multipath interference, or environmental dynamics. The proliferation of sensor-rich mobile devices presents a significant opportunity to enhance localization performance through sensor fusion and context-aware algorithms [23,133,134,135]. To improve robustness and adaptability, it is critical to move beyond reliance on single-technology solutions and instead adopt hybrid approaches that integrate complementary measurement techniques—such as combining radio frequency signals with inertial or magnetic field data. Such multimodal systems can mitigate the limitations of individual technologies, ensuring more reliable and accurate indoor positioning across diverse and dynamic environments.

2.Heterogeneity of Features

Device heterogeneity poses a significant challenge to the accuracy and reliability of IPSs. Variations in hardware components—such as antenna design, signal sensitivity, noise characteristics, and processing capabilities—lead to inconsistent signal measurements across different devices, even under identical environmental conditions. These discrepancies can result in substantial deviations in the received signal strength or other signal features, ultimately impairing localization accuracy [136,137]. This issue is particularly pronounced in fingerprint-based IPSs, where reference signatures are typically collected using specific devices as shown in Figure 5. During real-world deployment, users often carry heterogeneous mobile devices, causing mismatches between the live measurements and the stored fingerprint database. Even repeated measurements by the same device at the same location can yield varying signal profiles due to temporal and environmental dynamics, further complicating reliable position estimation [138]. Maintaining uniform hardware standards across all phases of data collection and usage is often impractical. In addition, heterogeneity can significantly degrade system performance, especially in dynamic indoor environments where signal propagation is affected by factors such as human movement, furniture rearrangement, and multipath interference. Addressing device heterogeneity remains a critical requirement for achieving robust and generalizable IPS deployments.

3.Measurement Techniques

Measuring a device’s location within an indoor environment is a challenging task. Indoor positioning requires accurate measurements of various signal features, such as RSS, CSI, TOA, Time of Delay (TOD), Time of Departure (TOD), TDOA, AOA, and so on as presented in Figure 5. Each of these techniques has its own strengths and weaknesses and are affected by different factors affecting indoor positioning accuracy [139,140,141,142]. All these measurement techniques are affected by various factors, including the propagation speed of the signal (which is affected by the medium), path loss, multipath, and obstacles. Thus, it is important to select the appropriate technique based on the requirements of the specific application and the environmental conditions. The selection process should consider the accuracy required, the coverage area, the cost, and the complexity of implementation.

4.Design Requirements

Figure 6 illustrates the multidimensional design requirements that underpin the performance of IPSs when considered as critical infrastructure. Central to the diagram is positioning performance, which is influenced by four interdependent design dimensions: space complexity, application/domain specificity, time complexity, and cost.

Space Complexity refers to the memory requirements and resource efficiency of positioning algorithms, which become particularly relevant in resource-constrained or embedded system environments.

Applications/Domains capture the contextual needs of various deployment scenarios (e.g., healthcare, logistics, emergency response), where spatial layout, user mobility, and operational priorities dictate system design.

Time Complexity addresses the computational demands of IPS algorithms, emphasizing the need for real-time or near-real-time localization in dynamic environments.

Cost encompasses both the capital and operational expenditures associated with IPS deployment, including hardware, software, and maintenance.

The figure emphasizes that no single factor can independently define system effectiveness. Rather, optimal IPS design involves balancing these factors based on domain-specific constraints. This holistic, multi-criteria perspective is essential for developing IPS solutions that are accurate, efficient, cost-effective, and context-aware.

5.Major Positioning Techniques

Numerous tactics used by IPSs can be methodically divided into three fundamental categories: positioning strategies, algorithmic approaches, and enabling technologies as depicted in Figure 7. By showing how physical hardware, signal processing techniques, and computational models all work together to influence system performance, scalability, and application scope, this classification makes it easier to comprehend IPS design coherently.

A.Algorithmic Methods

The algorithmic approach specifies how the system architecture calculates location estimation. Three main models are frequently used:(i)Centralized methods focus all localization calculations at a single processing unit, usually a server or cloud-based engine. Centralized models may face problems with latency, privacy, and scalability, especially in large-scale deployments or latency-sensitive applications, even though they are capable of complex inference and integration of heterogeneous data sources.(ii)Distributed methods delegate localization computation to specific mobile devices or nodes. By reducing communication overhead and facilitating localized inference, these systems improve scalability and robustness. However, because each node only has a limited amount of contextual information available, they might be less accurate.(iii)By using multi-node collaboration and successive refinement, iterative approaches combine the advantages of distributed and centralized schemes. This class includes methods like belief propagation, particle filtering, and message passing, which provide accuracy and flexibility at the expense of longer computation and convergence times.
B.Positioning Techniques

The fundamental signal processing approaches used to determine the spatial location of users or devices are referred to as positioning techniques. These can be broadly separated into two categories: range-based and range-free methods.

(i)Range-based techniques: These methods depend on the geometric properties of signal transmission and necessitate precise measurements of physical signal characteristics.○The TOA, TDOA, and RTT methods estimate distances from time delays between signal transmission and reception.○AOA makes use of directional data obtained by antenna arrays.○Using signal attenuation models, Received Signal Strength (RSS) calculates distance.

High-precision localization is usually possible with these techniques, but they require specialized hardware and strict synchronization.

(ii)Range-free techniques: Direct estimation of distance or angle is avoided by range-free techniques. Rather, they use network topology or pattern recognition to infer location: ○Pattern-matching algorithms (like k-NN and SVM) and pre-recorded signal characteristics (like RSSI vectors) are used in fingerprinting techniques.○Proximity-based models use the closest beacon or access point to determine the user’s location.○Using the network’s topological characteristics, connectivity and hop-count-based techniques (like DV-Hop) estimate location.

These methods may produce lower spatial resolution, but they are typically easier to implement and more resilient to hardware constraints.

C.Enabling Technologies

The third dimension concerns the hardware and physical infrastructure that make IPS deployment possible; these technologies are divided into two categories: device-based and device-free.

(i)Device-based technologies require an active device (such as a smartphone or BLE/UWB tag) to be carried by the user or object. These systems dominate consumer and commercial applications by taking advantage of the wide availability of mobile devices and their Wi-Fi, Bluetooth, or UWB compatibility.(ii)Device-free technologies passively determine a subject’s location without requiring them to carry any equipment. These consist of vibration/acoustic sensors, vision systems with depth or RGB cameras, and motion detection based on radio frequency. These methods are especially useful in situations involving non-cooperative tracking, security, and ambient assisted living. An organized and non-redundant framework for analyzing IPS design decisions is provided by this tripartite classification, which consists of approaches, techniques, and enabling technologies. It discusses the trade-offs between system accuracy, deployment complexity, energy efficiency, and cross-application domain adaptability. To achieve reliable, context-aware, and scalable indoor localization solutions, these dimensions must be in line with particular use-case requirements.

6.Domain-Specific Demands and Trade-Offs in IPS Deployment

Figure 8 illustrates the diverse application domains of IPSs, categorized based on their accuracy requirements, sensitivity, and power consumption constraints. The triangular structure emphasizes an inverse relationship between power consumption and required positioning accuracy across domains. At the top of the triangle—representing the most accuracy-sensitive and power-critical use cases—are applications in the medical sector, such as patient care, medical equipment tracking, pharmacy logistics, and emergency unit coordination. These domains demand real-time, high-precision localization with stringent reliability and privacy requirements.

Mid-tier domains, such as logistics and intelligent manufacturing, including asset inventory, material tracking, and industrial plant operations, represent a balanced trade-off between accuracy and power efficiency. These applications require moderate accuracy but must be scalable and robust under diverse operating conditions. At the base of the triangle are client-side applications like vehicle tracking, personnel monitoring, and general object positioning, where power efficiency is prioritized over precision, making them suitable for infrastructure-light or wearable systems. These domains are typically less sensitive and more tolerant to positional drift or delay. This classification underscores the heterogeneous and domain-specific performance demands placed on IPSs, reaffirming that no single solution can universally satisfy all application scenarios. The figure also highlights the critical need for tailored trade-off strategies in system design, guided by the operational sensitivity and functional requirements of each domain.

A concise overview of these interrelated challenges and trade-offs is presented in Table 1, highlighting their technical causes and practical implications for IPS design and deployment.

## 3. Experimental Results and Discussion

This section presents the key empirical findings derived from a series of real-world experiments conducted over a 25-month period, aimed at evaluating the long-term performance of an adaptive Wi-Fi fingerprinting-based indoor positioning system. The study focused on analyzing temporal signal strength variations and their impact on localization accuracy. RSSI data were collected across multiple time intervals and spatial locations to capture environmental dynamics. Initial data preprocessing was performed to extract meaningful features and ensure the robustness of subsequent analyses. The evaluation centered on four key aspects: (1) identifying the significance of signal features over time, (2) assessing the effect of sampling fluctuations on localization performance, (3) analyzing the dynamic behavior of indoor environments using a multidimensional metrics framework, and (4) benchmarking the proposed algorithm against standard machine learning models. The findings provide insights into the practical constraints, advantages, and design considerations essential for deploying resilient and scalable indoor localization systems.

(a)Wi-Fi Fingerprint-Based Indoor Location Estimation of Targets Utilizing Original Feature Spaces

To evaluate the impact of irrelevant or low-significance features on indoor localization accuracy, four training datasets were randomly selected from a specific month (Month 1), alongside five corresponding testing datasets from the same period. The proposed algorithm was benchmarked against multiple conventional algorithms to assess its robustness in long-term adaptive Wi-Fi fingerprinting-based localization under dynamic indoor conditions. Table 2, Table 3, Table 4 and Table 5 summarize the localization performance using the original feature space of 620 dimensions, with the MAE employed as the principal evaluation metric. The algorithms evaluated include Decision Tree (DT), K-Nearest Neighbors (KNN), Support Vector Classifier (SVC), Linear Regression (LR), Random Forest (RF), Gaussian Mixture Model (GMM), Multi-Layer Perceptron (MLP), and the proposed model.

Across all training and testing configurations, the proposed algorithm consistently achieved the lowest MAE values, indicating superior accuracy and robustness. Notably, performance degradation was observed in all algorithms—especially for testing samples 3 and 4—highlighting the influence of temporal signal variability. However, the proposed model maintained relatively stable performance, demonstrating resilience to signal fluctuations and variability in training data. These findings underscore the effectiveness of the proposed algorithm for long-term indoor localization and confirm its robustness in handling dynamic signal conditions typical of real-world Wi-Fi environments.

Table 3 presents a detailed evaluation of algorithm performance in Wi-Fi fingerprint-based indoor localization using two distinct training datasets (Samples 3 and 4). The analysis, conducted with a feature space of 620 dimensions, examines algorithmic robustness across five testing datasets under dynamic temporal signal conditions. The results clearly indicate that the proposed algorithm consistently outperforms all baseline models, achieving the lowest MAE across both training configurations. Despite the notable performance degradation observed in all algorithms when evaluated on testing samples 3 and 4—highlighting the impact of signal instability—the proposed algorithm demonstrates strong resilience to such fluctuations. These findings confirm the proposed method’s robustness in handling temporal variability and its effectiveness in maintaining reliable localization accuracy, even in scenarios characterized by significant signal dynamics.

Table 4 presents a comparative analysis of algorithm performance for Wi-Fi fingerprint-based indoor localization using two training datasets (Samples 1 and 2) and five testing datasets. The evaluation employs the RMSE as the performance metric and is conducted using the original feature space of 620 dimensions under dynamic temporal signal conditions. The results demonstrate that the proposed algorithm consistently yields lower RMSE values compared to baseline methods, confirming its superior localization performance. However, the RMSE values are notably higher than the corresponding MAE scores reported in Table 3. This discrepancy arises from RMSE’s sensitivity to large errors due to the squaring of residuals, which disproportionately penalizes outliers. Given the inherently dynamic nature of indoor environments—characterized by fluctuating signal strength and spatial variability—signal measurements tend to be dispersed, increasing the likelihood of outlier effects. Consequently, RMSE values provide a more stringent assessment of algorithm robustness under adverse conditions. Despite this, the proposed method maintains reliable performance, further underscoring its adaptability in complex indoor scenarios.

Table 5 unequivocally shows that the proposed algorithm consistently outperforms other algorithms in terms of achieving the lowest RMSE values. This superiority is evident in both training datasets (#3 and #4) collected during the first month. In contrast, all algorithms experience reduced accuracy when tested on datasets #3 and #4, indicating that dynamic signal fluctuations have a significant impact on localization performance in the indoor environment. These findings confirm that the proposed algorithm, along with the other algorithms studied, are highly affected by the changing indoor environment, leading to inflated negative results.

Figure 9 illustrates the robustness of various algorithms when trained on different samples collected during the first month, using the original 620-dimensional feature space. Performance is evaluated using two complementary error metrics: MAE and RMSE. The results show that RMSE values are consistently higher than MAE across all models, reflecting RMSE’s sensitivity to large errors due to the squaring of residuals. This characteristic makes RMSE particularly useful for penalizing significant deviations, which is relevant in dynamic indoor environments where signal measurements at fixed locations can be highly variable. Despite the elevated RMSE scores, the proposed algorithm demonstrates superior performance across both metrics, indicating strong generalization and robustness to temporal signal fluctuations. The consistently lower MAE values achieved by the proposed model further highlight its resilience to outliers, as MAE provides a more stable estimate of average localization error by treating all deviations equally without amplifying the impact of extreme values. Given the unpredictable nature of indoor signal propagation, both MAE and RMSE offer valuable but distinct perspectives on localization accuracy. MAE is preferable when robustness to outliers and interpretability are priorities, while RMSE is more appropriate for applications that require strong penalization of large localization errors. As such, employing both metrics provides a comprehensive assessment of algorithmic performance under real-world conditions.

## 4. Evaluation Criteria for IPSs

IPSs, particularly when deployed as components of critical infrastructure, must be evaluated through a comprehensive and multidimensional framework. Traditional metrics—such as accuracy and power consumption—remain central, but emerging application domains necessitate broader evaluation dimensions, including scalability, interoperability, and system resilience. This section outlines the primary evaluation criteria based on a synthesis of state-of-the-art literature and the trade-off analysis conducted in this study.

Accuracy

Accuracy is a core performance metric that reflects the spatial precision of an IPS. In safety-critical domains such as healthcare, emergency response, and industrial automation, high-accuracy localization is vital to ensure reliability and reduce operational risks. Technologies such as UWB and BLE using AoA methods have demonstrated sub-meter to centimeter-level accuracies in complex environments [66,75,90].

2.Energy Efficiency

Power consumption significantly impacts the viability of IPSs, especially in wearable or battery-powered configurations. Energy-efficient systems are necessary for sustained operations in healthcare and ambient assisted living applications, where devices may operate continuously for extended periods. Efforts such as the use of EnOcean energy-harvesting sensors and BLE-based localization systems aim to mitigate power constraints [71,77].

3.Reliability

Reliability pertains to the system’s ability to maintain stable performance under diverse conditions, including signal interference, multipath fading, or dynamic structural environments. Hybrid systems that combine BLE, Wi-Fi, and inertial sensors (IMUs) have been shown to improve robustness in real-time tracking and navigation tasks across multi-floor and crowded venues [40,51,69].

4.Scalability

Scalability determines how well an IPS can be expanded across large indoor spaces or high user densities without degradation in service quality. BLE mesh networks and Wi-Fi 6E solutions have enabled large-scale deployments in airports and smart factories with acceptable trade-offs in performance and cost [67,68,89].

5.Interoperability

The ability of an IPS to integrate with heterogeneous hardware, legacy systems, and external platforms is essential in real-world deployments. Interoperability facilitates seamless information exchange and system extensibility. Integration examples include IoT frameworks for healthcare [80] and standards-compliant BLE/UWB fusion in logistics and manufacturing [82,84].

6.Resilience

Resilience refers to the IPS’s capacity to withstand failures, signal fluctuations, or abrupt environmental changes. Multi-sensor fusion techniques and adaptive filtering (e.g., AI-enhanced BLE with Least Squares Estimation) have demonstrated improved continuity and adaptability in dynamic scenarios [75,78].

7.Privacy and Security

Privacy-preserving mechanisms are critical in sectors such as healthcare and public safety, where personal location data is sensitive. Proposed solutions include blockchain-based IPS frameworks (e.g., BILPAS) and group anonymity schemes for multi-user environments [100,101].

8.Cost and Ease of Deployment

The cost-effectiveness and deployment feasibility of IPSs directly impact their scalability and adoption. Wi-Fi fingerprinting, BLE beacon networks, and VLC represent low-cost alternatives that leverage existing infrastructure with minimal installation overhead [32,62,119].

In conclusion, evaluating IPSs requires balancing these criteria in alignment with domain-specific constraints and operational priorities. Trade-off analysis among accuracy, energy use, scalability, and privacy must inform design decisions to ensure system viability as critical infrastructure across diverse applications.

### 4.1. Trade-Off Analysis with GPS

While the GPS remains the de facto standard for outdoor localization due to its global coverage and high accuracy in open-sky environments, it faces significant limitations in indoor settings. GPS signals suffer from severe attenuation, multipath propagation, and NLOS conditions when obstructed by building structures, rendering them unreliable or unavailable for indoor positioning tasks [35]. IPSs address these limitations by leveraging alternative technologies such as Wi-Fi, BLE, UWB, and geomagnetic sensing to enable localization in GPS-denied environments. However, unlike GPS, IPSs present trade-offs that must be carefully managed depending on the deployment context. Compared to GPS, IPSs typically offer higher spatial resolution indoors, with technologies like UWB achieving centimeter-level accuracy [66]. However, this comes at the cost of infrastructure complexity and deployment effort, as IPSs often require environment-specific calibration, sensor installation, or radio map construction [36,73]. Additionally, energy efficiency and scalability remain critical challenges; many IPSs consume more power or struggle with large-scale deployment compared to the satellite-based GPS architecture [71,84]. In contrast to the unified architecture of GPS, IPSs are highly fragmented, relying on a variety of signal modalities, measurement techniques, and algorithms. This heterogeneity introduces interoperability issues, especially when integrating across different buildings or sectors [80,101].

Despite these challenges, IPSs offer several strategic advantages in mission-critical indoor applications. Unlike GPS, they can be customized for specific environments, support context-aware services, and ensure resilient tracking in dynamic and cluttered settings. Moreover, with advances in AI-driven signal processing and sensor fusion, IPSs are increasingly capable of adapting to environmental variability—features essential for positioning systems considered as critical infrastructure [75,77]. In this context, conducting a structured trade-off analysis is essential for evaluating IPSs as viable infrastructure solutions. Techniques such as SWOT (Strengths, Weaknesses, Opportunities, Threats) analysis, Multi-Criteria Decision Analysis (MCDA), and Cost–Benefit Analysis (CBA) can be used to assess key performance dimensions—including accuracy, reliability, cost, and deployment feasibility. These analytical frameworks guide decision-makers in identifying the optimal balance among competing objectives when implementing IPSs in complex environments. In summary, IPSs complement GPS by filling its operational void indoors, but their deployment involves trade-offs across accuracy, cost, energy use, and system complexity. A well-structured trade-off analysis is essential to determine the optimal balance between these factors, especially when IPSs are intended to serve as infrastructure-grade localization solutions.

### 4.2. Collaborative Indoor Positioning Systems

Collaborative Indoor Positioning Systems (CIPSS) represent an emerging paradigm in indoor localization that leverages the cooperation among multiple devices, sensors, or users to improve positioning accuracy, robustness, and scalability. Unlike traditional centralized systems, CIPSS distribute sensing, computation, and decision-making processes across participants, enabling enhanced localization performance in environments with dynamic signal conditions, obstructions, or limited infrastructure [53].

In decentralized or iterative configurations, each device actively contributes to the estimation of its position by exchanging signal measurements or inferred location data with neighboring nodes. These systems reduce dependence on static infrastructure, making them particularly suitable for applications in resource-constrained or infrastructure-free environments, such as elder care, emergency response, and mobile workforce tracking [41,53]. The integration of wearable sensors and BLE beacons in healthcare and assistive settings further exemplifies collaborative localization, where devices collectively track individuals in real-time without requiring extensive fixed installations [31,69]. For example, collaborative tracking in day care centers and hospitals allows caregivers to monitor multiple subjects with increased situational awareness and reduced manual oversight [69].

Moreover, the document emphasizes the need for infrastructure-free and collaborative IPS solutions to improve indoor location accessibility, particularly for visually impaired users and in public navigation services [32]. This aligns with the growing push for inclusive, low-cost, and adaptive localization systems. In summary, CIPSS offer significant advantages in terms of flexibility, scalability, and resilience, especially in scenarios where conventional IPSs face challenges due to infrastructure limitations or environmental dynamics. As indoor environments become increasingly complex and user-centric, collaborative positioning will play a critical role in the development of context-aware, intelligent indoor services.

## 5. Discussion and Conclusions

This paper critically examined IPSs through the lens of critical infrastructure, emphasizing the need for reliable, scalable, and context-aware localization technologies in safety-sensitive domains. By systematically reviewing technological architectures, application domains, and performance metrics, and by conducting empirical analysis of signal-based localization performance, several key insights and challenges emerged. The trade-off analysis confirmed that no single IPS technology satisfies all performance criteria. UWB systems offer high spatial accuracy but incur substantial deployment and energy costs, whereas BLE and Wi-Fi-based systems provide scalable and affordable alternatives with moderate accuracy but are more vulnerable to interference and environmental dynamics [66,71,90]. AI-enhanced fingerprinting, sensor fusion, and adaptive models can mitigate some of these limitations, but these improvements often come at the expense of increased computational complexity and calibration effort [72,75].

The empirical results from long-term Wi-Fi fingerprinting experiments revealed a strong sensitivity of localization performance to temporal signal variation, feature heterogeneity, and spatial inconsistencies. The proposed algorithm demonstrated robustness across time-varying datasets, consistently outperforming conventional models such as KNN, SVC, and Random Forest in terms of MAE and RMSE metrics. However, performance degradation in testing samples collected under fluctuating conditions reinforces the broader challenge of maintaining model generalizability in dynamic indoor environments. The literature synthesis identified a pressing need for more collaborative, hybrid, and infrastructure-light approaches. CIPSS, in particular, emerge as a promising direction for enabling decentralized, context-rich tracking across multiple agents or devices, especially in healthcare, logistics, and smart public infrastructure [31,32,69]. Meanwhile, application-specific demands—such as sub-meter precision in manufacturing, privacy in healthcare, and real-time adaptability in emergency response—further highlight the fragmented requirements that IPSs must satisfy [73,80,84]. Moreover, the discussion revealed critical research gaps that must be addressed for IPSs to mature into core components of digital infrastructure. These include:-Lack of standardization and interoperability across technologies and platforms [80,84];-Persistent challenges in privacy preservation and data security, particularly in continuous tracking applications [100,101];-The need for regulatory frameworks to govern IPS deployment in sensitive environments [46];-Trade-offs between energy efficiency and accuracy, especially for wearable and battery-limited systems [77,78].

In conclusion, the transformation of IPSs from auxiliary tools into critical infrastructure will depend on the development of resilient, adaptable, and ethically aligned systems. Future research should focus on hybrid architectures that combine AI, edge computing, and cross-sensor data fusion to balance performance with deployment feasibility. Additionally, establishing benchmarks for interoperability, security, and environmental adaptability will be essential for scaling IPSs across sectors. As the demand for precise indoor localization grows, IPSs must evolve to meet the technical, operational, and ethical imperatives of next-generation smart environments.

## 6. Conclusions and Future Directions

This paper has positioned IPSs as emerging components of critical infrastructure, moving beyond their traditional roles in commercial applications to serve safety-critical domains such as healthcare, industrial automation, public safety, and smart cities. By integrating a multidimensional evaluation framework—including empirical performance assessment, system design trade-offs, and application-specific requirements—this work provides a comprehensive lens through which IPS technologies can be analyzed, benchmarked, and improved. Key findings confirm that the viability of IPSs in mission-critical contexts is inherently shaped by complex trade-offs among accuracy, energy efficiency, scalability, cost, interoperability, and resilience. Technologies such as UWB, BLE, and Wi-Fi fingerprinting each offer distinct advantages but suffer from context-dependent limitations that preclude universal applicability. Furthermore, real-world experiments highlight those dynamic indoor environments introduce significant signal fluctuations, making long-term localization stability and model generalizability ongoing challenges. To ensure that IPSs evolve into reliable and adaptive critical infrastructure, the following future research directions are proposed:

Hybrid and Context-Aware Architectures: Future systems must adopt flexible architectures that integrate multiple sensing modalities—such as UWB, BLE, inertial sensors, and vision-based methods—tailored to application-specific constraints. These hybrid systems should dynamically adapt to environmental changes, user behavior, and infrastructural heterogeneity.

Collaborative and Decentralized Positioning Models: Leveraging collaborative indoor positioning frameworks, where multiple devices or agents share localization information, will be essential to improving robustness, especially in infrastructure-light or infrastructure-free settings such as emergency response or elderly care.

Standardization and Interoperability Frameworks: The lack of unified standards continues to hinder the scalability and cross-domain deployment of IPS solutions. Future work must focus on developing open, interoperable protocols and middleware layers that support seamless integration across heterogeneous platforms.

Energy-Efficient and Edge-Enabled Computation: To facilitate long-term deployment in mobile and wearable devices, IPS algorithms must be optimized for low-power execution. The incorporation of edge computing will enable local, real-time processing while preserving bandwidth and improving response latency.

Security, Privacy, and Ethical Governance: As IPSs increasingly track individuals and assets in sensitive domains, research must prioritize privacy-preserving mechanisms, secure communication protocols, and ethical frameworks to govern the responsible use of indoor location data.

Benchmarking and Real-World Validation: There is a pressing need for standardized evaluation datasets, testbeds, and performance benchmarks to support reproducibility and objective comparison of IPSs solutions under realistic deployment conditions. In conclusion, IPSs hold transformative potential across multiple domains, but realizing their promise as robust infrastructure components will require interdisciplinary innovation that bridges sensing technologies, artificial intelligence, system architecture, and ethical design. Future IPSs must not only be accurate and efficient but also adaptable, secure, and aligned with societal needs in increasingly complex indoor environments.

## Figures and Tables

**Figure 1 sensors-25-04914-f001:**
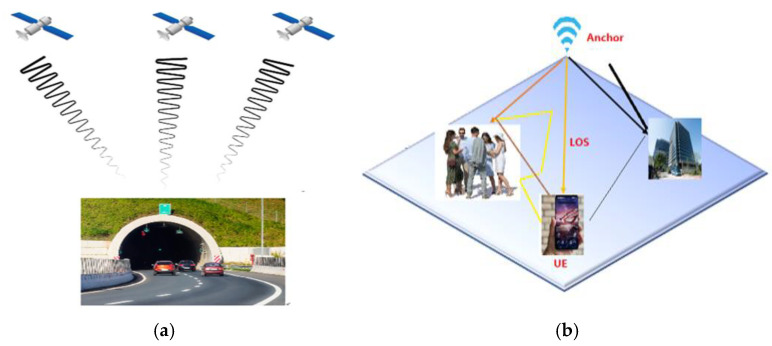
(**a**) GPS signal deterioration when the open sky is obscured. (**b**) Multipath effect for indoor scenario.

**Figure 2 sensors-25-04914-f002:**
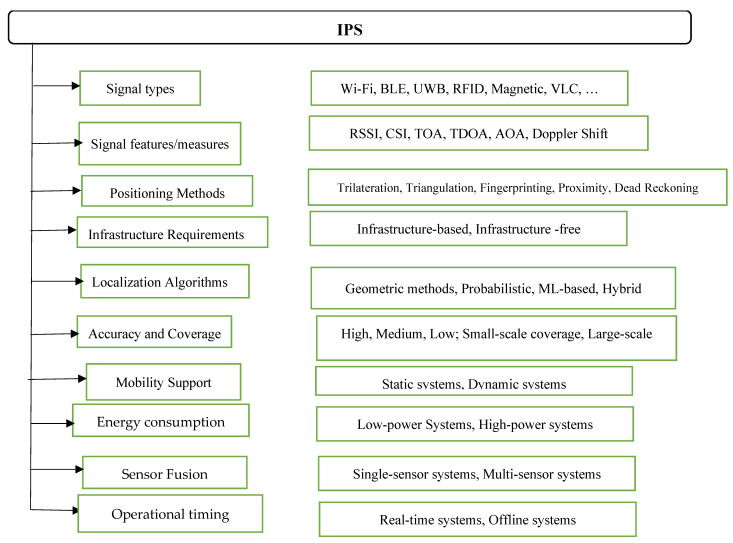
Core dimensions and taxonomic structure in the design and evaluation of IPS technologies.

**Figure 3 sensors-25-04914-f003:**
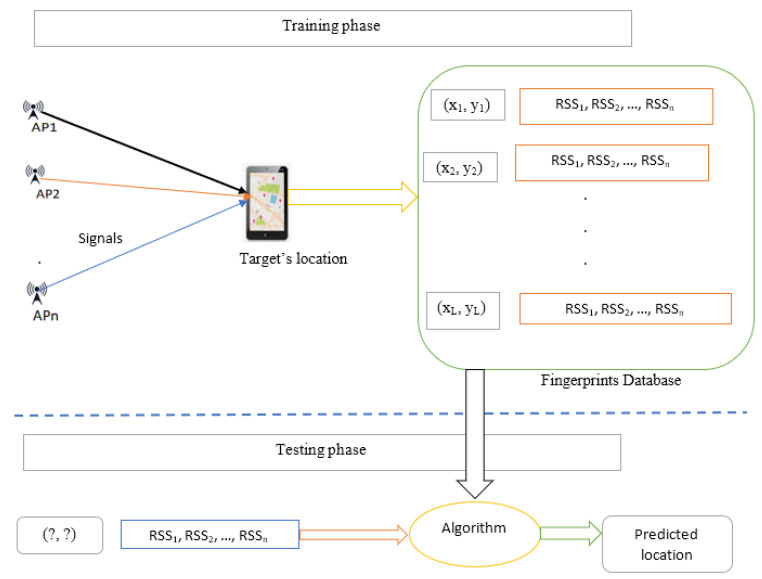
Overview of the Wi-Fi fingerprinting method for indoor localization.

**Figure 5 sensors-25-04914-f005:**
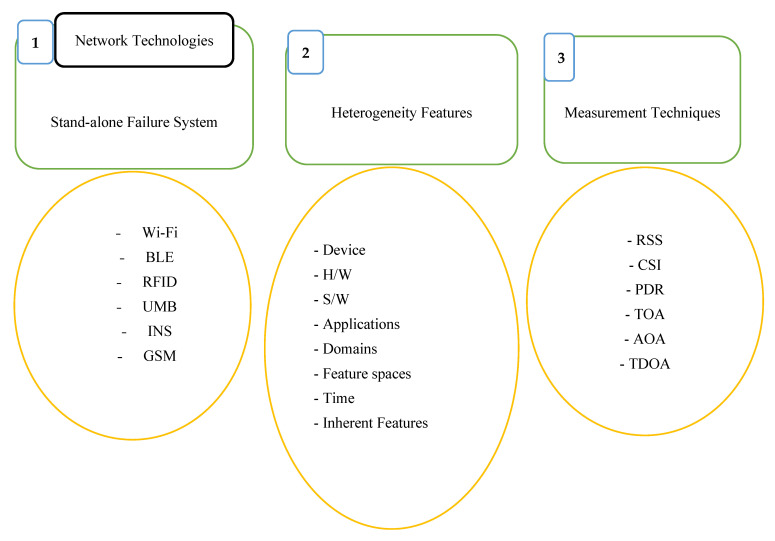
Challenges in selecting network technologies, heterogeneity features, and signal characteristics for IPSs.

**Figure 6 sensors-25-04914-f006:**
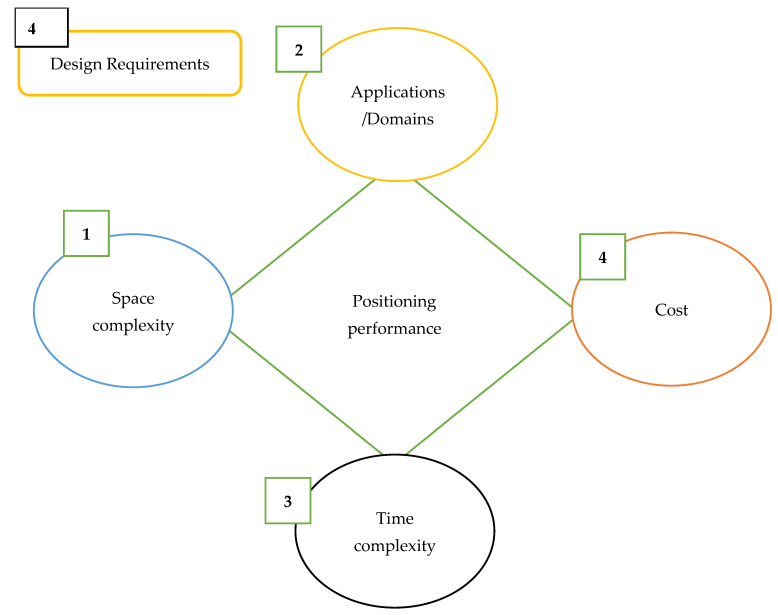
Multidimensional design requirements affecting for IPS performance.

**Figure 7 sensors-25-04914-f007:**
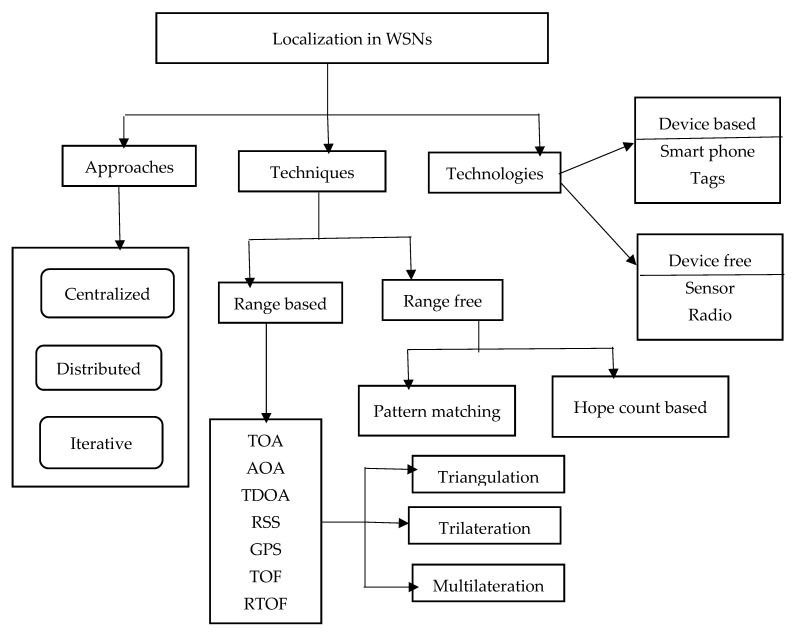
Classifications of major indoor localization techniques in WSNs.

**Figure 8 sensors-25-04914-f008:**
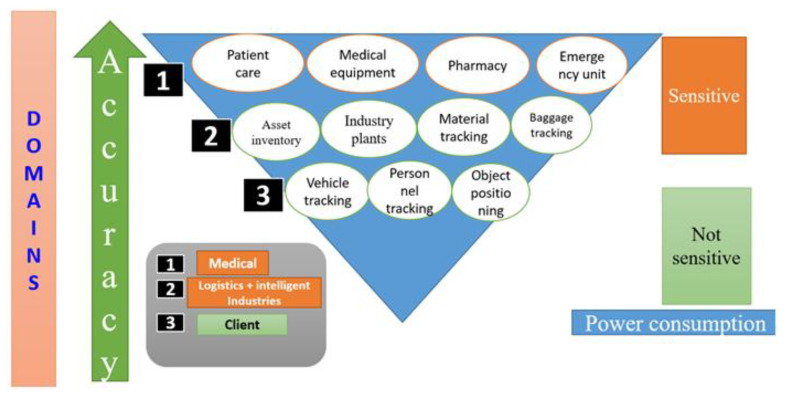
Categorization of IPS application sectors by required positioning precision and operational constraints.

**Figure 9 sensors-25-04914-f009:**
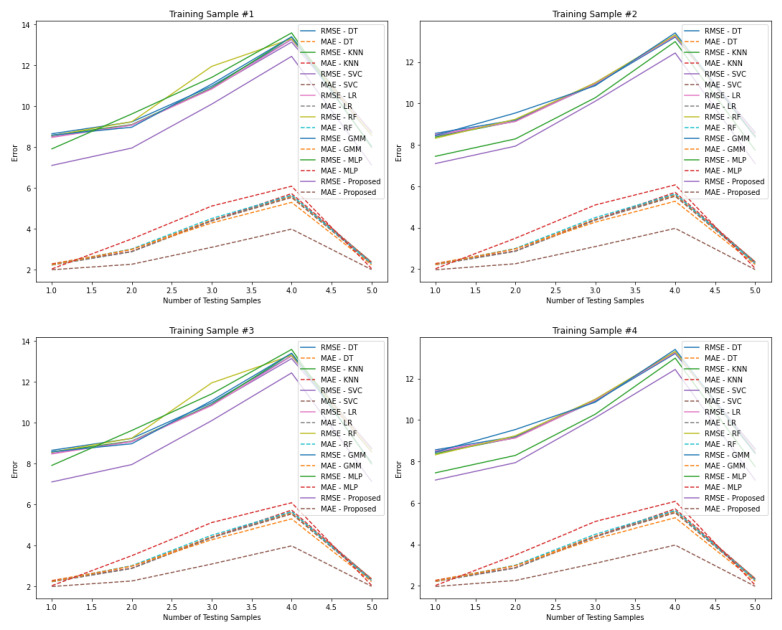
Robustness of algorithms evaluated against the effect of different training samples of a month 1.

**Table 1 sensors-25-04914-t001:** Summary of key challenges and trade-offs in indoor positioning systems.

Trade-off Dimension	Description	Cause/Constraint	Implications
Accuracy vs. Environmental Dynamics	High-accuracy systems degrade in NLOS or multipath environments	Signal fluctuation, occlusion, interference	Limits reliability in dynamic or cluttered indoor environments
Accuracy vs. Energy Efficiency	Improved precision increases computational and sensing load	AI models, multi-sensor fusion	Reduces battery life in wearables and mobile platforms
Scalability vs. Infrastructure	Greater area coverage often reduces localization precision	Sparse anchor nodes, infrastructure-light setups	Affects usability in large or multi-floor facilities
Model Generalizability vs. Signal Variability	Fingerprint models struggle with time-dependent or device-specific changes	Device heterogeneity, signal instability	Frequent re-calibration needed; impacts real-time tracking consistency
Privacy vs. Continuous Tracking	Real-time localization raises ethical and regulatory concerns	Data sensitivity in healthcare, public spaces	Requires privacy-preserving and secure system architectures
Standardization vs. Technological Diversity	Heterogeneous technologies lack unified standards	Vendor-specific protocols, diverse sensing modalities	Hinders cross-platform integration and interoperability

**Table 2 sensors-25-04914-t002:** Wi-Fi fingerprint-based indoor location estimation of targets utilizing original feature spaces of size 620 with dynamic temporal signal variations.

Criteria	Dataset Collected on Month 1									
	Training Dataset 1					Training Dataset 2				
	#Testing Samples					#Testing Samples				
	#MAE (in m)									
	1	2	3	4	5	1	2	3	4	5
DT	2.31	2.97	4.41	5.57	2.33	2.30	2.98	4.38	5.57	2.27
KNN	2.26	2.88	4.38	5.71	2.34	2.19	2.94	4.45	5.67	2.18
SVC	2.22	2.87	4.38	5.52	2.28	2.20	2.94	4.49	5.61	2.21
LR	2.21	2.86	4.36	5.62	2.36	2.23	2.92	4.42	5.62	2.27
RF	2.22	3.00	4.49	5.63	2.23	2.21	2.98	4.49	5.68	2.20
GMM	2.23	3.11	4.27	5.67	2.06	2.08	3.21	4.51	5.67	2.08
MLP	2.02	3.49	5.10	6.07	2.04	1.93	2.52	4.18	5.49	1.99
Proposed	1.98	2.25	3.08	3.97	1.99	1.97	2.26	3.09	3.97	1.98

**Table 3 sensors-25-04914-t003:** Wi-Fi fingerprint-based indoor location estimation of targets utilizing original feature spaces of size 620 with dynamic temporal signal variations.

Criteria	Dataset Collected on Month 1									
	Training Dataset 3					Training Dataset 4				
	#Testing Samples					#Testing Samples				
	#MAE (in m)									
	1	2	3	4	5	1	2	3	4	5
DT	2.21	3.05	4.49	5.72	2.24	2.24	3.05	4.39	5.72	2.28
KNN	2.22	2.98	4.44	5.69	2.26	2.17	2.99	4.48	5.84	2.19
SVC	2.18	2.94	4.44	5.61	2.27	2.23	2.99	4.48	5.78	2.22
LR	2.21	2.91	4.43	5.66	2.28	2.29	3.02	4.43	5.76	2.29
RF	2.16	3.01	4.52	5.71	2.20	2.15	3.08	4.55	5.82	2.24
GMM	2.35	2.98	4.27	5.73	2.36	2.20	3.05	4.41	5.67	2.29
MLP	1.87	3.48	5.14	6.02	1.88	1.99	2.72	4.35	5.65	1.97
Proposed	1.96	2.27	3.08	3.99	1.98	1.98	2.26	3.08	3.98	1.98

**Table 4 sensors-25-04914-t004:** Wi-fi fingerprint-based indoor location estimation of targets utilizing original feature spaces of size 620 with dynamic temporal signal variations.

Criteria	Dataset Collected on Month 1									
	Training Dataset 1					Training Dataset 2				
	#Testing Samples					#Testing Samples				
	#RMSE (in m)									
	1	2	3	4	5	1	2	3	4	5
DT	8.64	9.22	10.97	13.26	8.74	8.56	9.20	10.91	13.21	8.61
KNN	8.56	9.09	10.89	13.38	8.71	8.38	9.15	10.96	13.30	8.38
SVC	8.49	9.06	10.86	13.13	8.57	8.44	9.14	11.00	13.25	8.45
LR	8.46	9.07	10.83	13.26	8.73	8.51	9.13	10.95	13.27	8.57
RF	8.55	9.23	10.94	13.29	8.58	8.32	9.24	11.00	13.31	8.44
GMM	8.57	8.96	10.93	13.39	8.03	8.46	9.54	10.86	13.41	8.41
MLP	7.90	9.61	11.40	13.58	7.97	7.45	8.29	10.28	12.99	7.75
Proposed	7.10	7.91	10.09	12.43	7.12	7.10	7.94	10.11	12.44	7.10

**Table 5 sensors-25-04914-t005:** Wi-Fi fingerprint-based indoor location estimation of targets utilizing original feature spaces of size 620 with dynamic temporal signal variations.

Criteria	Dataset Collected on Month 1									
	Training Dataset 3					Training Dataset 4				
	#Testing Samples					#Testing Samples				
	#RMSE (in m)									
	1	2	3	4	5	1	2	3	4	5
DT	8.42	9.35	10.98	13.43	8.52	8.50	9.30	10.93	13.40	8.67
KNN	8.44	9.19	10.92	13.32	8.55	8.40	9.20	11.00	13.50	8.43
SVC	8.41	9.14	10.89	13.26	8.58	8.52	9.22	11.03	13.45	8.49
LR	8.46	9.08	10.87	13.34	8.56	8.61	9.27	10.98	13.42	8.61
RF	8.36	9.24	10.95	13.44	8.36	8.41	9.27	11.00	13.45	8.36
GMM	8.12	9.00	10.85	13.26	8.95	8.76	9.29	11.00	13.05	8.31
MLP	7.16	9.30	11.24	13.28	7.16	7.79	8.75	10.63	13.16	7.74
Proposed	7.05	7.97	10.12	12.47	7.07	7.05	7.91	10.06	12.5	7.03

## Data Availability

The dataset used for this study are available upon request to the corresponding author.

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
