# Peer review of "Indoor Positioning Systems as Critical Infrastructure: An Assessment for Enhanced Location-Based Services"

_sensors, 2025, doi:10.3390/s25164914_

Round 1

Reviewer 1 Report

Comments and Suggestions for Authors

This paper introduces a multi-dimensional trade-off framework that blends technical and operational metrics—including interoperability, privacy, and resilience, for enhanced location based services. The proposed method adds practical value to real-life applications.

Here are some questions:

  1. Wi-Fi is foacsed in this article. How about other important IPS technologies, like BLE or UWB. Can you comment on them?
  2. Incoporating domain-specific trials such as healthcare examples would be better for demonstrating the practical value. Even using some simple setup mimicking this environment is also a good step.
  3. privacy issue was mentioned many times but there are no investigation on it. Considering implementing at least a basic privacy-preserving protocol and test it would be nice.

Author Response

Review #1:

Comments and Suggestions for Authors

This paper introduces a multi-dimensional trade-off framework that blends technical and operational metrics—including interoperability, privacy, and resilience, for enhanced location based services. The proposed method adds practical value to real-life applications.

Here are some questions:

Question #1: Wi-Fi is foacsed in this article. How about other important IPS technologies, like BLE or UWB. Can you comment on them?

Response #1: Thank you for your insightful comment. While this paper primarily focuses on Wi-Fi-based indoor positioning for experimentation, we critically review several other important IPS technologies, including BLE and UWB, as part of the state-of-the-art. Although we don't delve into the exhaustive details of each technology, we have provided a comprehensive critical overview of their principles and relevance within the broader IPS landscape to establish context for our focused study.

Question #2: Incoporating domain-specific trials such as healthcare examples would be better for demonstrating the practical value. Even using some simple setup mimicking this environment is also a good step.

Response #2:  We agree with the reviewer's insightful comment regarding the value of incorporating domain-specific trials, such as healthcare examples, to better demonstrate the practical utility of our work. This is indeed an important consideration. Our current scope was delimited to verifying the foundational implementation. However, we fully agree that this is a crucial and impactful direction for our future work.

Question #3: privacy issue was mentioned many times but there are no investigation on it. Considering implementing at least a basic privacy-preserving protocol and test it would be nice.

Response #3: We appreciate the reviewer's constructive and insightful comment regarding the importance of implementing privacy-preserving protocols. We entirely agree that this is a crucial consideration for IPSs. Our current study primarily focused on verifying the foundational implementation and core performance aspects within the scope of our proposed evaluation framework. While we critically explored the state-of-the-art concerning privacy throughout various related works to substantiate our framework, the practical implementation and testing of a privacy-preserving protocol were delimited to the scope of this particular study. Nevertheless, we accept the fact that incorporating such protocols is a crucial and impactful direction to be considered in a separate future research work.

Thank you!

Reviewer 2 Report

Comments and Suggestions for Authors

The paper under review thoroughly explored indoor positioning systems in various scenarios, including critical infrastructure, and emphasized the need for reliable and context-aware localization technologies in safety-sensitive domains.

Besides the overview, the empirical results from long-term Wi-Fi fingerprinting experiments are revealed and proposed algorithm demonstrated robustness across time-varying datasets.

The text of the article is written in clear language, consistent and logical. The conclusions are reasoned and substantiated

The presented manuscript can hardly be called a breakthrough work, however, it presents the current state of the art in the area under study in a comprehensive manner and may be of interest to readers engaged in research in the field of indoor network positioning.

Author Response

Review #2:

Comments and Suggestions for Authors

The paper under review thoroughly explored indoor positioning systems in various scenarios, including critical infrastructure, and emphasized the need for reliable and context-aware localization technologies in safety-sensitive domains.

Besides the overview, the empirical results from long-term Wi-Fi fingerprinting experiments are revealed and proposed algorithm demonstrated robustness across time-varying datasets.

The text of the article is written in clear language, consistent and logical. The conclusions are reasoned and substantiated

The presented manuscript can hardly be called a breakthrough work; however, it presents the current state of the art in the area under study in a comprehensive manner and may be of interest to readers engaged in research in the field of indoor network positioning.

Response:

We sincerely thank the reviewer for their thorough assessment and highly encouraging feedback. We are pleased that the paper effectively explored indoor positioning systems across various scenarios, including critical infrastructure, and successfully highlighted the need for reliable and context-aware localization technologies in safety-sensitive domains. We also appreciate the positive remarks on our empirical results from long-term Wi-Fi fingerprinting experiments, and we're glad the proposed algorithm's robustness across time-varying datasets was clearly demonstrated. The reviewer's commendation of the article's clear, consistent, and logical writing, as well as our reasoned and substantiated conclusions, is particularly gratifying. Besides, we agree with the assessment that it comprehensively presents the current state of the art in the field. Our aim was indeed to offer a valuable resource for researchers engaged in indoor network positioning, and we are delighted that the reviewer believes it achieves this goal.

Thank you!

Reviewer 3 Report

Comments and Suggestions for Authors

The author proposes a novel evaluation framework that combines traditional performance indicators with emerging requirements such as system energy consumption and privacy protection, and discusses a large number of key challenges such as achieving privacy protection and evaluation standardization in practical applications. The research objective of the article is of great value. However, the article has problems such as inconsistent main ideas throughout the text, content jumps and incoherence. Especially in the experimental part, it does not verify the series of evaluation frameworks proposed in the article, but conducts experimental verification of the influencing factors of WiFi fingerprint positioning accuracy. Specifically as follows:

  1. Three main positioning algorithms are introduced in Chapter 2 Overview of IPSs. Among them, scene analysis was not introduced; In the “2.2 related work”, three fundamental goals were proposed: ubiquity, seamlessness, and intelligence but the subsequent content did not correspond well to these three contents.
  2. Approaches, techniques, and technologies in “2.3.5 Major Positioning Techniques”. Where is the difference in the division? There is a lot of repetition and redundancy, and the content description is also incomplete. It is suggested to reorganize it.
  3. The “Experimental Results and Discussion” discuss the four key factors of WiFi fingerprint indoor positioning. This content has no corresponding relationship with the evaluation framework mentioned earlier. Many Trade-off Dimensions are mentioned in the second chapter of the text. Should the experiment start from these perspectives?
  4. The author's sudden insertion of WIFI fingerprint positioning in the Overview of IPSs seems very abrupt. The previous text does not mention the introduction of related fingerprint positioning, and the subsequent text does not correspond with wifi fingerprint either, which appears very redundant. Whether the subsection "Trade-off Analysis with GPS" in the "Evaluation Criteria for IPSs" is inconsistent with the topic of this article. This article studies indoor positioning, and whether the trade-off analysis with GPS here is redundant.
  5. Inconsistent terminology. Such as approximation and Proximity based estimation; Bluetooth and BLE. Meanwhile, the Signal Measurement Methods and positioning methods in the text have been used interchangeably. If the author refers to the signal measurement method, then the positioning method should not be listed in Figure 2. If the author refers to the positioning method, then the RSSI therein is a measurement signal, not a positioning method. The author needs to sort out the concepts.
  6. The pictures are not clear.

Author Response

Review #3:

Comments and Suggestions for Authors

The author proposes a novel evaluation framework that combines traditional performance indicators with emerging requirements such as system energy consumption and privacy protection, and discusses a large number of key challenges such as achieving privacy protection and evaluation standardization in practical applications. The research objective of the article is of great value. However, the article has problems such as inconsistent main ideas throughout the text, content jumps and incoherence. Especially in the experimental part, it does not verify the series of evaluation frameworks proposed in the article, but conducts experimental verification of the influencing factors of WiFi fingerprint positioning accuracy. Specifically as follows:

Question #1: Three main positioning algorithms are introduced in Chapter 2 Overview of IPSs. Among them, scene analysis was not introduced;

Response #1: Thank you for the insightful comment. In response, we acknowledge the omission and have now included a concise introduction to scene analysis in Chapter 2. Specifically, we describe it as a positioning algorithm that relies on collecting signal features from the environment to build a reference database (offline phase) and then matching real-time measurements to this database (online phase), commonly used in fingerprinting-based methods. Thus, the section has been revised accordingly to ensure completeness and clarity and marked in yellow.

Question #2: In the “2.2 related work”, three fundamental goals were proposed: ubiquity, seamlessness, and intelligence but the subsequent content did not correspond well to these three contents.

Response #2: We appreciate the reviewer's constructive observation regarding the alignment between the proposed fundamental goals (ubiquity, seamlessness, and intelligence) and the subsequent content in Section 2.2. We have revised this section to explicitly connect the discussed related work to these three core objectives, ensuring a clearer and more coherent presentation of how existing research addresses or contributes to each goal. This will involve rephrasing some descriptions and potentially adding specific sentences to highlight these connections. Thus, for detail we have highlighted in yellow in the revised manuscript.

Question #3: Approaches, techniques, and technologies in “2.3.5 Major Positioning Techniques”. Where is the difference in the division? There is a lot of repetition and redundancy, and the content description is also incomplete. It is suggested to reorganize it.

Response #3: We agree with the reviewer's assessment regarding the organization and redundancy in Section 2.3.5, 'Major Positioning Techniques. We have thoroughly reorganized this section to eliminate redundancy, streamline the presentation, and provide a more comprehensive yet concise overview of major positioning techniques. Our revision has focused on a clearer categorization and more detailed, distinct explanations for each technique, ensuring a more logical flow and improved clarity. For the changes we made, we have highlighted in yellow in the revised manuscript.

Question #4: The “Experimental Results and Discussion” discuss the four key factors of WiFi fingerprint indoor positioning. This content has no corresponding relationship with the evaluation framework mentioned earlier. Many Trade-off Dimensions are mentioned in the second chapter of the text. Should the experiment start from these perspectives?

Response #4: We appreciate the reviewer's valuable comment regarding the connection between our "Experimental Results and Discussion" and the evaluation framework and trade-off dimensions previously introduced. We agree that our experimental evaluation should more directly address these aspects. Our study specifically focused on analyzing temporal signal strength variations and their impact on localization accuracy, critically considering the dynamic nature of indoor environments. In our manuscript, we've explicitly linked our experimental results to the trade-off dimensions discussed in Chapter 2. Our evaluation was scoped around four key aspects that inherently address these dimensions:

  • Understanding Data Volatility: We identified the significance of signal features over time.
  • Addressing Deployment Constraints: We assessed the effect of sampling fluctuations on localization performance.
  • Real-world Environment Analysis: We analyzed the dynamic behavior of indoor environments using a multidimensional metrics framework that prioritizes real-world settings over generic metrics.
  • Practical Performance Benchmarking: We benchmarked our proposed algorithm against standard machine learning models.

These findings collectively offer insights into the practical constraints, advantages, and design considerations essential for deploying resilient and scalable indoor localization systems, directly reflecting the trade-offs discussed earlier. We acknowledge that not all factors mentioned in our comprehensive evaluation framework were covered in this specific experimental study. Each of these factors can constitute independent research avenues, and while we provided a critical analysis of the state-of-the-art around them, this particular work centered on the temporal aspects of signal strength.

 Question #5: The author's sudden insertion of WIFI fingerprint positioning in the Overview of IPSs seems very abrupt. The previous text does not mention the introduction of related fingerprint positioning, and the subsequent text does not correspond with wifi fingerprint either, which appears very redundant. Whether the subsection "Trade-off Analysis with GPS" in the "Evaluation Criteria for IPSs" is inconsistent with the topic of this article. This article studies indoor positioning, and whether the trade-off analysis with GPS here is redundant.

Response #5: Thank you for the thoughtful comment. We have carefully addressed all the concerns raised in the revised manuscript. Thus, the detail change made can be seen highlighted in yellow in the revised manuscript on its respective section.

Question #6: Inconsistent terminology. Such as approximation and Proximity based estimation; Bluetooth and BLE. Meanwhile, the Signal Measurement Methods and positioning methods in the text have been used interchangeably. If the author refers to the signal measurement method, then the positioning method should not be listed in Figure 2. If the author refers to the positioning method, then the RSSI therein is a measurement signal, not a positioning method. The author needs to sort out the concepts.

Response #6: Thank you for the thoughtful comment. In the revised manuscript, we have carefully addressed all the concerns raised. Thus, the detail change made can be seen highlighted in yellow in the revised manuscript on its respective section.

Question #7: The pictures are not clear.

Response #7: Thank you for your comment. In the revised manuscript, we have revised to enhance the quality of the figures.

Thank you!

Round 2

Reviewer 1 Report

Comments and Suggestions for Authors

The author answered the questions from their perspective by emphasizing the specific scope of the article. However, I did not find additional changes made to the revised manuscript regarding these questions.

Author Response

Thank you for your message and for guiding us through the review process. We appreciate the reviewers’ constructive feedback and the opportunity to provide clarifications.

Please find below our point-by-point response addressing each comment from Reviewer #1:

Reviewer #1:

Comment 1: Wi-Fi is foacsed in this article. How about other important IPS technologies, like BLE or UWB. Can you comment on them?

Response:
Thank you for this insightful question. While our study emphasizes Wi-Fi fingerprinting for experimentation, we have included a critical review of other relevant IPS technologies, including BLE and UWB, in the manuscript. The decision to focus on Wi-Fi was guided by its widespread availability, cost-effectiveness, and suitability for large-scale indoor environments. We agree that BLE and UWB offer important features, such as energy efficiency and high accuracy, respectively. However, detailed implementation and evaluation of these technologies were beyond the scope of this work.

Comment 2: Incoporating domain-specific trials such as healthcare examples would be better for demonstrating the practical value. Even using some simple setup mimicking this environment is also a good step.

Response: 
We fully agree with the reviewer that domain-specific use cases, such as healthcare environments, can enhance the practical relevance of the work. Our current implementation was designed to establish a generalizable evaluation framework and to validate the foundational aspects of the proposed method. While we did not include specific healthcare scenarios in this study, we view this as an excellent direction for future research and are planning to extend our trials into domain-specific environments, including healthcare and smart building applications.

Comment 3: privacy issue was mentioned many times but there are no investigation on it. Considering implementing at least a basic privacy-preserving protocol and test it would be nice.

Response:
We appreciate this important observation. Indeed, privacy is a critical concern in IPS research, and we highlighted this in our manuscript to emphasize its importance. While our framework incorporated privacy as one of the multidimensional evaluation criteria, we acknowledge that the practical implementation of a privacy-preserving protocol was not part of the current experimental scope. We believe integrating and testing such protocols is vital and plan to explore this in a dedicated follow-up study, particularly by assessing how various privacy-preserving techniques affect localization performance and system trade-offs.

We hope these responses clarify the rationale behind our design choices and demonstrate our intention to address the reviewers’ suggestions in future work.

Thank you again for your time and consideration. We look forward to the academic editor’s feedback.

Best regards,
Tesfay G.